# Compact IF2 allows initiator tRNA accommodation into the P site and gates the ribosome to elongation

Ritwika S. Basu[1], Michael B. Sherman[2,3] & Matthieu G. Gagnon ⬤ [1,2,3,4 ✉]

During translation initiation, initiation factor 2 (IF2) holds initiator transfer RNA (fMet-tRNA$_i^{fMet}$) in a specific orientation in the peptidyl (P) site of the ribosome. Upon subunit joining IF2 hydrolyzes GTP and, concomitant with inorganic phosphate (P$_i$) release, changes conformation facilitating fMet-tRNA$_i^{fMet}$ accommodation into the P site and transition of the 70 S ribosome initiation complex (70S-IC) to an elongation-competent ribosome. The mechanism by which IF2 separates from initiator tRNA at the end of translation initiation remains elusive. Here, we report cryo-electron microscopy (cryo-EM) structures of the 70S-IC from *Pseudomonas aeruginosa* bound to compact IF2-GDP and initiator tRNA. Relative to GTP-bound IF2, rotation of the switch 2 α-helix in the G-domain bound to GDP unlocks a cascade of large-domain movements in IF2 that propagate to the distal tRNA-binding domain C2. The C2-domain relocates 35 angstroms away from tRNA, explaining how IF2 makes way for fMet-tRNA$_i^{fMet}$ accommodation into the P site. Our findings provide the basis by which IF2 gates the ribosome to the elongation phase.

[1] Department of Microbiology and Immunology, University of Texas Medical Branch, Galveston, TX 77555, USA. [2] Department of Biochemistry and Molecular Biology, University of Texas Medical Branch, Galveston, TX 77555, USA. [3] Sealy Center for Structural Biology and Molecular Biophysics, University of Texas Medical Branch, Galveston, TX 77555, USA. [4] Institute for Human Infections and Immunity, University of Texas Medical Branch, Galveston, TX 77555, USA. ✉email: magagnon@utmb.edu

Translation initiation is the rate-limiting step of protein biosynthesis[1]. In bacteria, this fundamental step in gene expression is regulated by initiation factors 1 (IF1), 2 (IF2), and 3 (IF3). IF2 is the guanosine triphosphatase (GTPase) that guides the entry of the formyl-methionine initiator tRNA (fMet-tRNA$_i^{fMet}$) into the peptidyl (P) site of the ribosome. In the small subunit (30 S) initiation complex, IF2-GTP holds fMet-tRNA$_i^{fMet}$ in a specific orientation and provides anchor points that accelerate joining of the large subunit (50 S), resulting in the formation of the 70 S initiation complex (70S-IC)[2–6]. Ribosome-stimulated GTP hydrolysis in IF2 and inorganic phosphate (P$_i$) release is accompanied by a series of conformational changes in IF2 and the ribosome, leading to the accommodation of fMet-tRNA$_i^{fMet}$ into the P site and IF2 dissociation from the ribosome[7–16]. These events are essential for the maturation of the 70S-IC into an elongation competent ribosome[5].

Initiation factor IF2 is the largest translation factor in *Escherichia coli* with an N-terminal region of variable length across bacterial species and a conserved C-terminal region that encompasses four domains (Supplementary Fig. 1a, b). Solution and crystal structures of IF2 suggest that the G-domain (domain I) and domain II together form a super-domain while domain III (C1) and the fMet-tRNA$_i^{fMet}$-binding domain IV (C2) appear to remain flexible[17–20]. Cryo-electron microscopy (cryo-EM) reconstructions of initiation complexes have shown that GTP-bound IF2 adopts an extended conformation with its β-barrel distal domain C2 interacting with the A76-fMet moiety of initiator tRNA in the p/I state[2,3,6,9,10,21] and that IF2 induces a rotated conformation of the 30 S subunit in the 70S-IC[2,9,10,21,22]. Kinetic studies[8,12,14] and time-resolved cryo-EM[10] revealed that upon GTP hydrolysis by IF2 in the 70S-IC after subunit association, initiator tRNA is released from the C2-domain, accommodates into the P site (p/P state), and the ribosome takes the non-rotated conformation. Single-molecule FRET studies suggest that the nucleotide-bound to the G-domain allosterically modulates the position of the terminal domain C2 relative to that of dI-dIII and initiator tRNA[7]. Thus, it is presumed that IF2 changes conformation during the last step of translation initiation, facilitating the accommodation of fMet-tRNA$_i^{fMet}$ into the P site and dissociation of IF2 from the ribosome. However, IF2 bound to the ribosome has only been visualized in the extended conformation interacting with fMet-tRNA$_i^{fMet}$ positioned in the p/I state, which fails to explain how domains in IF2 rearrange to grant access of initiator tRNA to the P site and commit the ribosome to elongation.

We report cryo-EM structures of the *Pseudomonas aeruginosa* 70S-IC bound to IF2, mRNA, and initiator fMet-tRNA$_i^{fMet}$. The structures reveal the hitherto unseen compact state of IF2-GDP bound to the ribosome. The large conformational changes in IF2 originate from the relatively small-scale rearrangement of the switch 2 (sw2) α-helix in the G-domain and propagate to the distal tRNA-binding domain C2, which collapses onto domain C1 allowing initiator tRNA accommodation into the P site. The structures visualize the interfaces between domains in compact IF2 and the ribosome, providing a detailed view of the IF2-mediated transition of the 70S-IC to a ribosome ready to begin the elongation phase.

## Results

### Cryo-EM structures of the 70 S ribosome initiation complex.
To capture IF2 in different conformation bound to the 70S-IC, we used the fact that IF2 co-purifies with GDP (Supplementary Fig. 2) as observed previously[19]. We hypothesized that the addition of the non-hydrolysable GTP analog β,γ-methylene-guanosine 5'-triphosphate (GDPCP) to the complex could yield at least two populations of the 70S-IC, one bound to IF2-GDPCP and the other to IF2-GDP. Cryo-EM and three-dimensional (3D) classification revealed two main populations of ribosome complexes bound to IF2 and featuring the non-rotated (class average I) and rotated (class average II) 30 S subunit (Supplementary Fig. 3; Methods). The two class averages appear to be reminiscent of the low-resolution cryo-EM reconstructions of the *Thermus thermophilus* ribosome bound to IF2, in which the 30 S subunit is rotated in the 70S-IC bound to IF2-GDPCP and non-rotated upon binding IF2-GDP[9]. Non-rotated ribosomes from class average I feature initiator tRNA that is fully accommodated into the P site. Compared to the extended IF2 conformation featured in all of the previous ribosome initiation complex structures[2,3,6,9,10,21–23], IF2 has a globular and compact shape in the non-rotated 70S-IC with its distal C2-domain collapsed onto domain C1 (Fig. 1a, b). Rotated ribosomes from class average II contain extended IF2 with its C2-domain interacting with fMet-tRNA$_i^{fMet}$ in the p/PI state (Fig. 1c, d), as previously reported in the *E. coli* ribosome initiation complexes (Supplementary Fig. 4)[2,3,6,10,21,22].

Despite the overall high-resolution of the reconstructions (2.6 – 2.9 Å) (Supplementary Fig. 5a, b), the EM density corresponding to domains C1 and C2 in compact IF2 was fragmented, possibly owing to high flexibility of this region in IF2[7,17,18,22,23]. Similarly, the uL11-stalk comprising the GTP-activating center (GAC) in the 50 S subunit displays high flexibility in both the non-rotated (class average I) and rotated (class average II) ribosome (Supplementary Fig. 6). To resolve the conformational heterogeneity of IF2 and the ribosome, we further classified particles based on 3D variability analysis focused on IF2 and the uL11-stalk region (Supplementary Fig. 3). This approach, combined with focused refinement of IF2 and the surrounding ribosomal elements[24–26], allowed to visualize the long-sought compact IF2-GDP in the 70S-IC.

### Compact IF2-GDP is bound to the non-rotated 70S-IC.
The non-rotated ribosome particles were sorted on the basis of the presence of continuous EM density for compact IF2, which yielded a subset of 21,480 particles that resulted in the 2.9-Å reconstruction of the 70S-IC with solid density for *P. aeruginosa* IF2 (structure I-A). Focused refinement with signal subtraction produced a density map of compact IF2 with a nominal resolution of 3.4 Å, with several regions extending to ~3.0-Å resolution (Supplementary Figs. 3 and 5c; Supplementary Table 1). The quality of this map resolves most of the side chains in IF2, allows to trace the linkers between domains, and visualizes the GDP nucleotide bound in the G-domain (Fig. 2a; Supplementary Fig. 7).

In structure I-A, the 30 S subunit is rotated by ~1.4° suggesting that compact IF2 exists in the 70S-IC that is essentially non-rotated. Correspondingly, the initiator tRNA is fully accommodated into the P site with its CCA-end base paired with the 23 S rRNA P-loop and the fMet residue positioned in the peptidyl transferase center (PTC) (Supplementary Fig. 8a), mimicking a ribosome complex ready to enter the elongation step. Compared to extended IF2, large-scale movements of domain C1 and the distal tRNA-binding domain C2 have occurred in compact IF2 (Fig. 3a). Domain C1 flips by 116° around the long axis of α-helix H8 of IF2 out of the inter-subunit cleft to relocate 29 Å away under the uL11-stalk of the 50 S subunit, losing all interactions with domain II (Fig. 3a, c, e; Supplementary Fig. 9). New interfaces form with the G- and C2-domains, and with the SRL and H43 of 23 S rRNA (Figs. 4a–c, e, 5a–b, e). Instead of α-helix H9 in domain C1 interacting with the G-domain in the extended conformation, it is α-helix H10 on the opposite side of C1 that

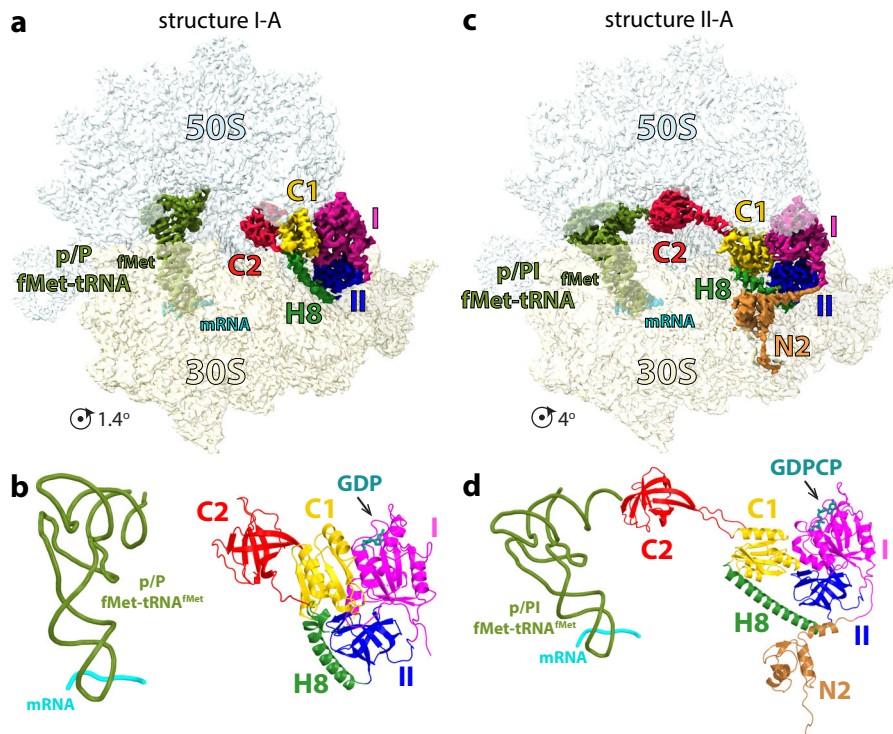

**Fig. 1 Cryo-EM structures of the 70 S ribosome initiation complex bound to IF2-GDP and IF2-GDPCP. a** Cryo-EM map of the *Pseudomonas aeruginosa* 70S-IC bound to compact IF2-GDP in structure I-A. The 50 S and 30 S subunits are shown in light blue and yellow, respectively. The initiator tRNA is green, mRNA is cyan, and domains of IF2 are assigned distinct colors. **b** Model of compact IF2-GDP in structure I-A. Relative to extended IF2 (**c**, **d**), the distal tRNA-binding domain C2 (red) collapses onto the core domains of compact IF2. **c** Cryo-EM map of the 70S-IC bound to extended IF2-GDPCP in structure II-A. **d** Model of extended IF2-GDPCP in structure II-A. The C2-domain (red) interacts with the fMet moiety of initiator tRNA bound in the p/PI state. The rotation angle of the 30 S subunit in the 70S-IC relative to the non-rotated *P. aeruginosa* 70 S ribosome[44] is indicated under each map (**a**, **c**). The N2 subdomain (brown) of IF2 is only visible in the rotated 70S-IC (**c**, **d**).

does this in compact IF2 (Fig. 3e, f). The large movement and rotation of domain C1 pulls domain C2 away from initiator tRNA through the inter-domain linker encompassing residues 735–742 that is bent by 180° between H12 and the C2-domain (Fig. 4c; Supplementary Movie 1). Domain C2 retracts by 35 Å and rotates by 152° to relocate near 23 S rRNA helices H89-H91-H92, and the SRL (H95) (Figs. 3a, d, 5a–c). The C2-domain docks against domain C1 (Figs. 1b and 4b; Supplementary Fig. 7), and Arg[742] in the C1-C2 linker region engages in electrostatic interactions with a negative patch lining one side of helix H12 in domain C1 (Fig. 4c). The relocation of domain C1 in compact IF2 is facilitated by the flexible helix H8, which tilts by 25° and sharply bends by 90° at the apical region around Leu[625] (Figs. 3b and 4d).

The absence of density for the γ-phosphate or inorganic phosphate ($P_i$) in the nucleotide-binding pocket of compact IF2 is consistent with bound GDP (Fig. 2a). This is additionally supported by the observation that the sw1 region is disordered and sw2 rearranged, moving the catalytic His[399] more than 8 Å away from the nucleotide binding pocket and 13 Å from the SRL (Figs. 2a and 3g). In the extended conformation of IF2-GDPCP, both switch regions contact domain C1 (Fig. 3f; Supplementary Fig. 10), contributing to its rigidity. Upon GTP hydrolysis and $P_i$ release, sw1 becomes disordered and loses interactions with domain C1 while sw2 is remodeled. This releases structural restraints on domain C1 which is then free to relocate under the uL11-stalk in compact IF2 (Fig. 5a, e). In IF2-GDP, the sw2-α-helix in the G-domain rotates by 65° providing a stabilizing interface for helix H10 in the C1-domain (Figs. 3e, g and 4e). The position of the sw2-α-helix in IF2-GDPCP is not compatible with the location of domain C1 under the uL11-stalk in IF2-GDP

(Fig. 3h), indicating that GTP hydrolysis and $P_i$ release must occur for IF2 to transition to the compact conformation on the ribosome. In compact IF2, domains are held together mostly by stacking interactions, with domain C1 sandwiched between the G- and C2-domains, together surrounding the SRL of 23 S rRNA (Fig. 5a, b; Supplementary Fig. 7). The β-strand Val[672]-(Gly)$_3$-Val[676] and helix H10 in domain C1 form the interface with the sw2-α-helix and H4 in the G-domain (Fig. 4e), while on the other side of domain C1, the flat surface formed by residues Asn[698]-Val[699]-Arg[700] stacks with β-hairpin residues Leu[782], Asp[785]-Val[786]-Val[787] in domain C2 (Fig. 4b).

The interactions between compact IF2 and the ribosome are different from those seen with extended IF2. Domain C2 resides in the vicinity of 23 S rRNA helices H89-H91-H92 and H95 (SRL), and of the C-terminal tip of r-protein uL6, together forming one side of the accommodation corridor for the incoming aminoacyl-tRNA (Fig. 5a). In this position, domain C2 interacts weakly with the ribosome. Residue Ser[818] is within hydrogen bonding distance from the 2'OH group of A2458 (*E. coli* A2471) and the ribose O4' atom of G2459 (G2472) in H89, and no interaction is seen with H91-H92 and uL6 (Fig. 5c). The carboxyl group of Asp[784], located in the apical loop of β-hairpin 781–788 of domain C2, interacts with the exocyclic amino group of G2648 (G2661) in the SRL (Fig. 5b). This is in contrast with the location of domain C2 in extended IF2-GDPCP on the rotated ribosome which, in addition to interacting with the CCA-end of fMet-tRNA$_i$[fMet], also contacts H89, H71 and ribosomal protein uL16 (Supplementary Fig. 11a, c–e). Similarly, the location of domain C1 in compact IF2 is strikingly different from that in extended IF2 on the rotated ribosome, in which the C1-domain

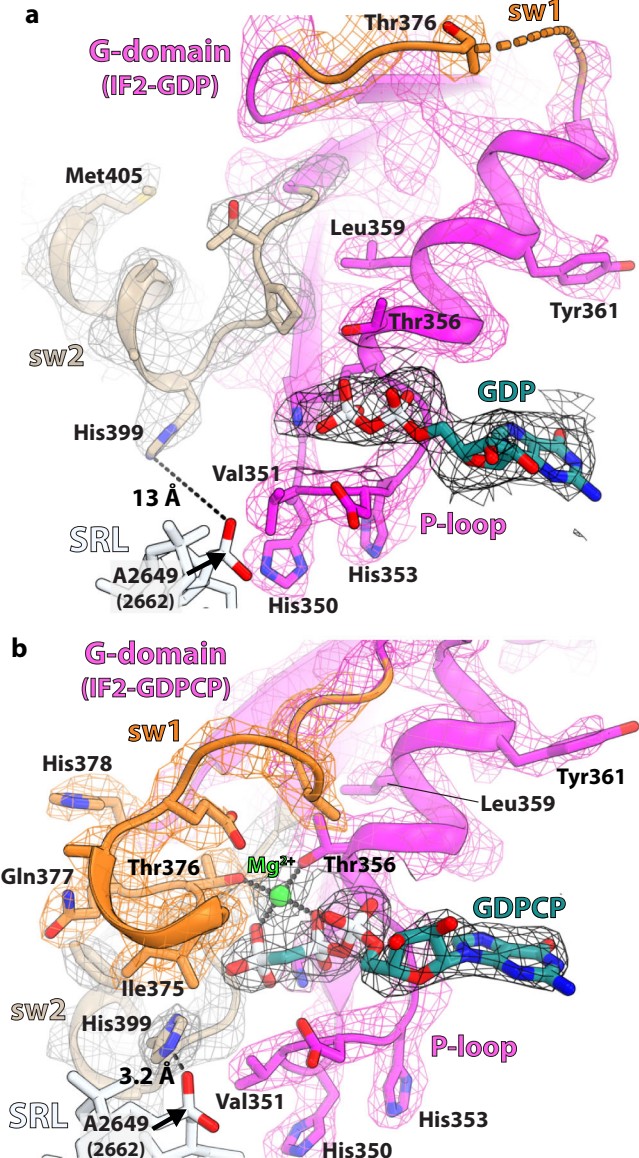

**Fig. 2 Nucleotide-binding pocket in the G-domain of IF2. a** The EM density of the GDP nucleotide and switch 2 (sw2) region in compact IF2 is shown as black and gray mesh, respectively, sw1 as orange mesh and the P-loop as pink mesh. The lack of density for sw1 (orange) indicates that it is disordered. The catalytic His[399] residue in sw2 of IF2-GDP is oriented away from GDP and locates 13 Å from the phosphate oxygen of A2649 (*E. coli* A2662) in the SRL. **b** In extended IF2, the catalytic His[399] is directed toward the γ-phosphate of GDPCP through the hydrophobic gate consisting of residues Val[351] of the P-loop and Ile[375] of sw1. In this position, His[399] is positioned to interact with the phosphate oxygen of A2649 (A2662) in the SRL. Residues Thr[376] of sw1 and Thr[356] of the P-loop, together with the β- and γ-phosphate oxygen atoms of GDPCP, coordinate a magnesium ion (green).

locates in the inter-subunit cleft formed by 16 S rRNA helix h5, and r-proteins uS12 and uL14 (Supplementary Figs. 9 and 11a–b). Under the uL11-stalk, domain C1 in compact IF2 docks against the GTP-activating center (GAC) helix H43 and the SRL (Fig. 5a; Supplementary Fig. 9). The apical nucleotide A2647 (A2660) in the SRL stacks with Val[676] and Gly[677] in domain C1, while the conserved Tyr[723] in H12 of domain C1, which in extended IF2 is within interaction distance with the catalytic His[399] (Supplementary Fig. 10e), stacks with nucleotide A1057 (A1067) in H43 of

GAC in compact IF2 (Fig. 5b, e). The interactions between domain II of compact IF2 and the 30 S subunit are preserved in structure I-A (Supplementary Fig. 12). The main chain of residues Gly[528]-Arg[529] and the side chain of Arg[558] stack with nucleotides A55 (A55) and U362 (U368), respectively, located in the body of the 30 S subunit, in keeping with the importance of A55 during protein synthesis (Fig. 5d)[27]. Likewise, contacts between the G-domain and the ribosome are largely unchanged, except that the rotation of the sw2-helix in compact IF2 disrupts its interaction with the SRL (Fig. 2a).

**Mechanism of IF2-GDP dissociation from the non-rotated ribosome.** Variability analysis of the non-rotated ribosome particles also revealed structure I-B with a nominal resolution of 2.9 Å (Supplementary Figs. 3 and 5a). Focused refinement shows IF2-GDP bound in a noticeably different position in structure I-B (Fig. 6a, b). While the inter-domain contacts in IF2 remain largely unchanged, the G-domain rotates by ~20° inward around the SRL resulting in ~10 Å-displacement at the outskirt of the G-domain, ~7 Å for domain C1, and ~5 Å for domain II (Fig. 6a). The displacement of IF2 brings helix H6 in the G-domain proximal to the N-terminal domain of uL11 and domain C2 within interaction distance from H92 of 23 S rRNA (Fig. 6a, c, d). Helices H43 and H44 in the uL11-stalk shift by ~7 Å inward along with domain C1 (Fig. 6a; Supplementary Fig. 6b). The movement of compact IF2 in structure I-B increases the gap between the 50 S subunit and the G-domain by ~4 Å around the SRL, and the displacement of domain II away from the 30 S subunit combined with further back-rotation of the 30 S subunit, from ~1.4° in structure I-A to ~0.7° in structure I-B, eliminate interactions with the small subunit (Fig. 6a, b). Correspondingly, the total buried surface area by compact IF2-GDP is reduced from ~2616 Å² in structure I-A to ~2409 Å² in structure I-B, consistent with weaker interactions between ribosome I-B and IF2-GDP primed to dissociate from the 70S-IC[9,15]. The absence of interaction between domain II and the 30 S subunit also explains the scattered low-resolution density attributable to domain II, indicating that it becomes highly dynamic in the inter-subunit space in structure I-B (Supplementary Fig. 5c). These observations provide insights into how IF2-GDP dissociates from the ribosome. As the 30 S subunit rotates back, compact IF2 releases its hold from the ribosome in a stepwise fashion; the G-domain peels off from the SRL and domain II dissociates from the ribosome before the other domains C1 and C2, similar to the proposed dissociation mechanism of EF-Tu and EF-G from the elongating ribosome[28,29].

**Dynamics of the uL11-stalk in the rotated 70S-IC.** In the rotated ribosome with extended IF2 (class average II), the density of the uL11-region is more scattered than that of the ribosome and IF2, suggesting that the uL11-stalk samples different conformations. Focused 3D variability analysis around extended IF2 and the uL11-stalk revealed structures II-A and II-B with global resolutions of 2.7 Å and 2.6 Å, respectively (Supplementary Figs. 3 and 5b). In II-A and II-B, the 30 S subunit is rotated by ~4°, representing a semi-rotated ribosome as previously reported[10,14,21]. The 70S-IC in structures II-A and II-B displays well-defined density for extended IF2 and GDPCP in the G-domain (Fig. 2b), differing only by the position of the uL11-stalk. The tip of H43 is positioned 25 Å outward in structure II-A relative to that in II-B (Supplementary Fig. 6a), and despite the large movement of the uL11-stalk, it does not perturb the conformation of extended IF2-GDPCP stably bound to the ribosome. With the stalk in the outward position, the N-domain of uL11 interacts with the G-domain of IF2 (Supplementary Fig. 6a), which may assist in positioning protein bL12 such that it promotes rapid subunit

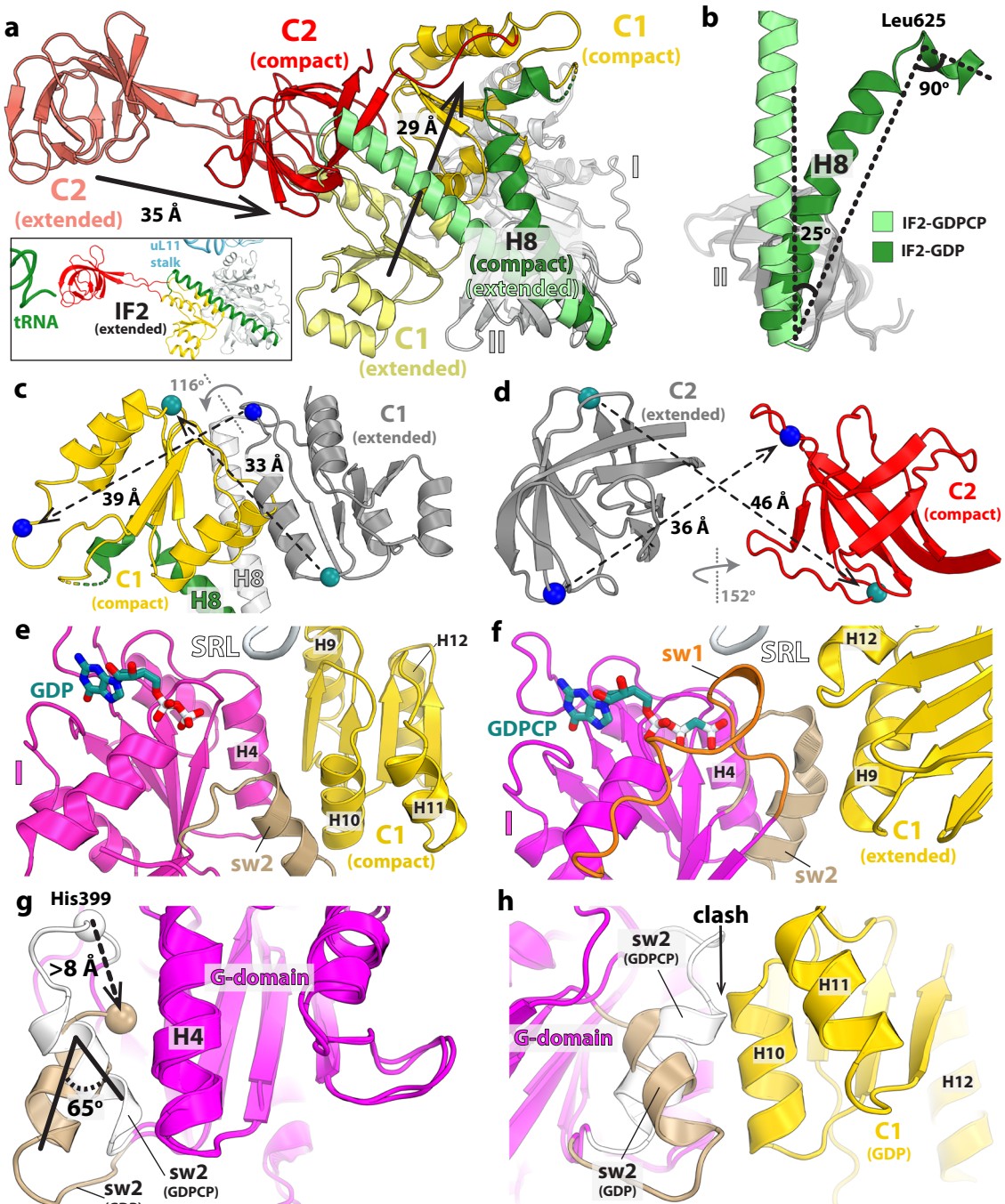

**Fig. 3 Structural rearrangements in IF2. a** Superimposition of the G-domain (gray) in structures I-A and II-A reveals domain rearrangements between extended (faded colors) and compact (darker colors) IF2. Domains C1 and C2 move by 29 Å and 35 Å, respectively. Inset: Overview of IF2 bound to the 70S-IC viewed in the same orientation. **b** Alignment of domain II in IF2 shows that helix H8 (green) bends by 25° and turns sharply by 90° around residue Leu[625] to accommodate the displacement of domain C1. **c**, **d** Movement of domains C1 and C2 between the extended (gray) and compact (red or yellow) IF2 is accompanied by a rotation of 116° for domain C1 and 152° for domain C2. The displacement of corresponding Cα atoms is indicated. **e**, **f** Position of domain C1 relative to the G-domain in compact (**e**) and extended (**f**) IF2. In IF2-GDP, helix H10 of the C1 domain (yellow) faces the sw2-α-helix (tan) in the G-domain (magenta) (**e**), while in IF2-GDPCP helices H9 and H12 from the opposite side of domain C1 face the G-domain (**f**). **g** Rotation of the sw2-helical region by 65° in IF2-GDP forms the new interface with domain C1, displacing the catalytic His[399] residue by more than 8 Å away from the nucleotide-binding pocket. **h** The conformation of sw2 in extended IF2-GDPCP (white) is not compatible with the location of domain C1 (yellow) in compact IF2-GDP.

association[30] and GTP hydrolysis by IF2[31]. The close proximity of the N-domain of uL11 and the G-domain of IF2 in structure II-A is in agreement with previous FRET studies reporting that upon 70S-IC formation, the G-domain of IF2 moves closer to the uL11-NTD both in the presence of GTP or a non-hydrolysable GTP analogue[32].

In structures I-A and I-B with compact IF2-GDP, the uL11-stalk is in the inward position similar to that observed in structure II-B (Fig. 6a; Supplementary Fig. 6). This allows nucleotide A1057 (A1067) at the tip of H43 to form a stacking interaction with conserved Tyr[723] in domain C1 that moved from the inter-subunit space in extended IF2 (Supplementary Fig. 11a–b) to its

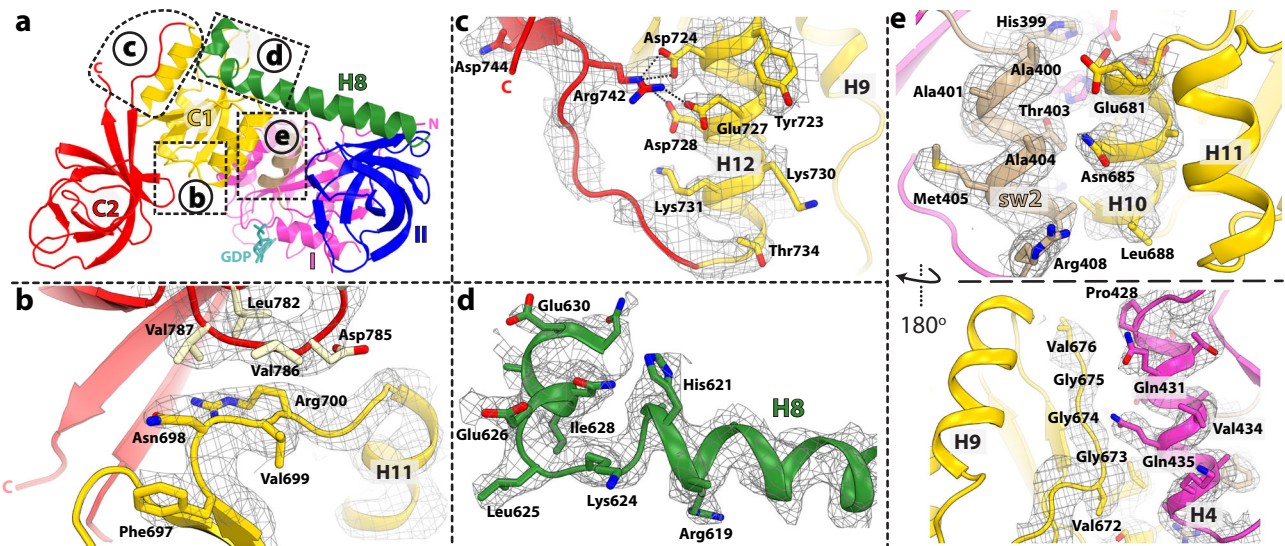

**Fig. 4 Interactions between domains in compact IF2. a** Overview of IF2-GDP with inset boxes indicating close-up views. **b** Residues Asn[698], Val[699], and Arg[700] in domain C1 (yellow) form a flat surface that stacks with β-hairpin residues Leu[782], Asp[785], Val[786], and Val[787] in domain C2 (red). EM density is shown as gray mesh. **c** Residues 735–742 form the linker between domains C1 (yellow) and C2 (red), which turns sharply by 180°. The last residue of the linker region (Arg[742]) interacts with the negative patch lining one side of helix H12 in domain C1. **d** EM density (gray mesh) of helix H8 (green) around residue Leu[625]. The 90°−turn of helix H8 is required to accommodate the new location of domain C1 in IF2-GDP. **e** Two views of the interface between the G-domain and domain C1. The β-strand Val[672]-(Gly)[3]-Val[676] (lower panel) and helix H10 (upper panel) in domain C1 form the interface with H4 (lower panel) and the sw2-α-helix (upper panel) in the G-domain, respectively.

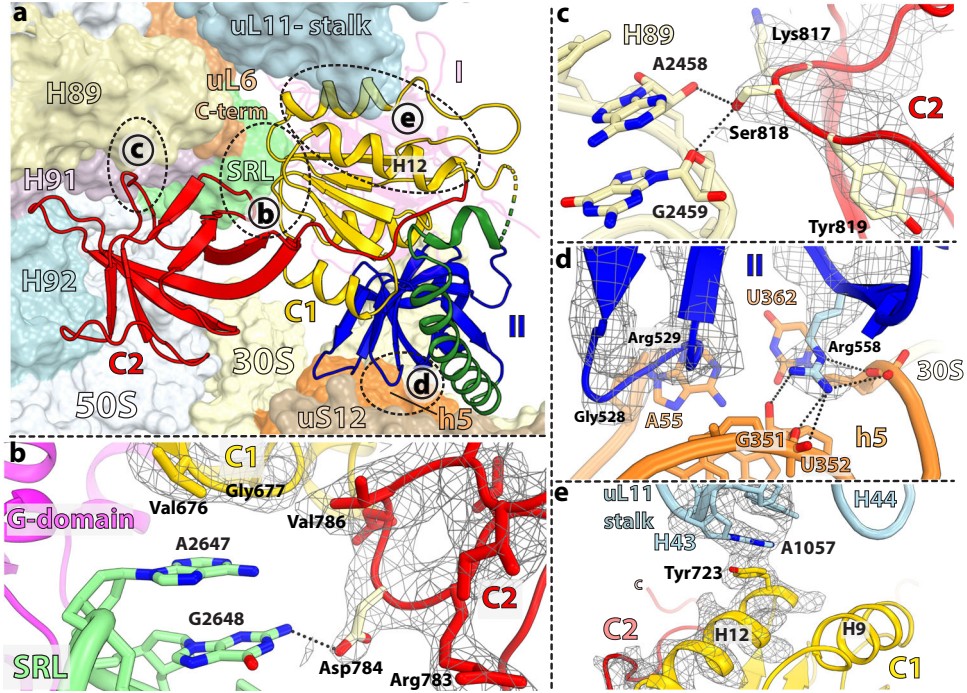

**Fig. 5 Interactions between compact IF2 and the ribosome. a** Overview of compact IF2-GDP bound to the 70 S ribosome. Domain C1 (yellow) locates under the uL11-stalk (light blue) while domain C2 (red) is disposed along the aminoacyl-tRNA accommodation corridor in the vicinity of 23 S rRNA helices H89 (pale yellow), H91 (light pink), H92 (light blue), and the SRL (pale green), and the C-terminal of r-protein uL6 (light orange). Domain II (blue) interacts with helix h5 of 16 S rRNA (orange). **b**–**e** Magnified views of the insets shown in **a**. **b** Domain C1 is sandwiched between the G- and C2-domains, together surrounding the SRL of 23 S rRNA (pale green). Residue Asp[784] in domain C2 (red) interacts with G2648 (*E. coli* G2661) in the SRL. Residues Val[676] and Gly[677] in domain C1 (yellow) form a flat surface that stacks with the apical nucleotide A2647 (A2660) in the SRL. EM density shown as gray mesh. **c** Residue Ser[818] in domain C2 (red) is within hydrogen bonding distance to nucleotides A2458 (A2471) and G2459 (G2472) in 23 S rRNA helix H89 (pale yellow). **d** The main chain of residues Gly[528] and Arg[529], and the side chain of Arg[558] in domain II (blue) stack with base pair A55-U362 (A55-U368) in helix h5 of 16 S rRNA (orange). **e** The conserved Tyr[723] in helix H12 of domain C1 (yellow) stacks with nucleotide A1057 (A1067) in 23 S rRNA helix H43 of the uL11-stalk.

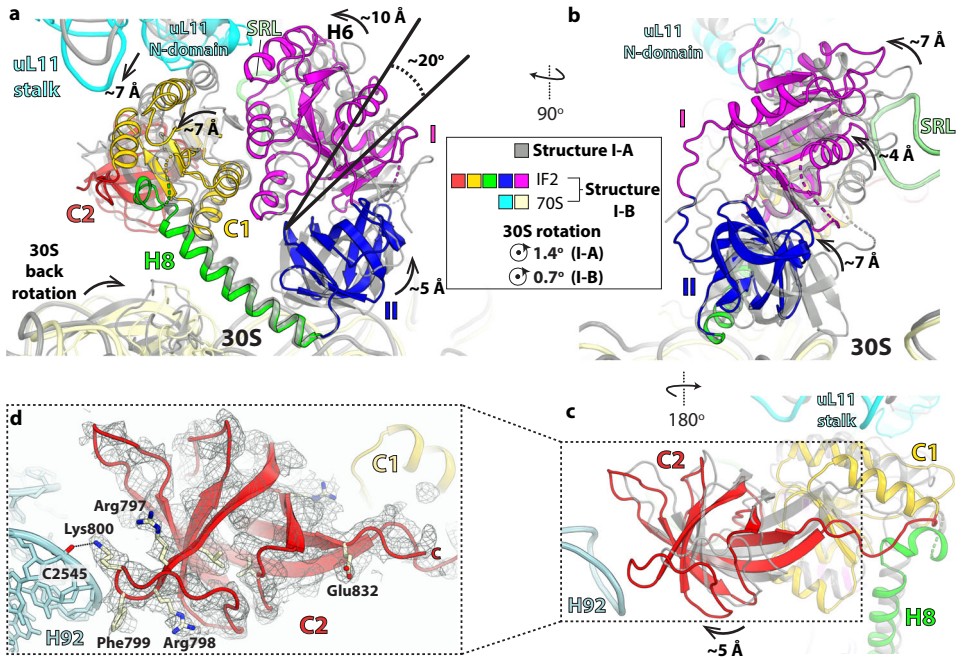

**Fig. 6 Compact IF2 is loosely bound to the ribosome in structure I-B. a** Alignment of the 23 S rRNA in structures I-A and I-B shows that domain I (G-domain) rotates inward by ~20° around the SRL in structure I-B (colored domains) relative to structure I-A (gray). The center of rotation was taken around residue Gly[589] in domain II. The maximum displacement at the outskirt of domain I is ~10 Å, ~7 Å for domain C1, and ~5 Å for domain II. Concomitantly, the uL11-stalk moves inward by ~7 Å together with domain C1. In structure I-B, helix H6 in the G-domain of compact IF2 is proximal to the N-domain of ribosomal protein uL11. The displacement of domain II away from the 30 S subunit in structure I-B, combined with the reverse rotation of the 30 S subunit between structure I-A (~1.4°) and I-B (~0.7°), weakens the binding of IF2-GDP to the ribosome in structure I-B. **b** Orthogonal view of panel **a** showing that in structure I-B the G-domain moves farther from the SRL by ~4 Å, which may facilitate IF2-GDP dissociation from the ribosome. **c** In structure I-B, the rotation of IF2-GDP around the SRL brings domain C2 (red) near 23 S rRNA helix H92. **d** EM density map of domain C2 in structure I-B shows that residues Phe[799] and Lys[800] form a complementary surface with H92. We note that in extended IF2 residue Phe[799] interacts with the fMet residue attached to fMet-tRNA$_i^{fMet}$ in structures II-A and II-B (see Supplementary Fig. 8b).

location under the uL11-stalk in compact IF2 (Fig. 6a, e; Supplementary Fig. 9).

**The N2 sub-domain anchors extended IF2 on the rotated ribosome**. The N-domain of IF2 is known to promote stable joining of the 30 S and 50 S subunits to form the 70S-IC[22]. IF2 in the extended conformation in the *P. aeruginosa* 70S-IC interacts with the ribosome and A76-fMet of p/PI-tRNA as seen in the *E. coli* 70S-IC (Supplementary Figs. 4, 6a, 8b and 11)[10,21]. We observe additional density that emerges from the N-termini of the G-domain of extended IF2 and reaches the shoulder domain of the 30 S subunit (Figs. 1c, Fig. 7a), paralleling previous tethered cleavage data of rRNA[33] and cryo-EM reconstructions[2,21]. The region of the N-domain of IF2 proximal to the G-domain that interacts with the 30 S subunit was previously called the N/G1 domain in *T. thermophilus* IF2[22,34], and shares high sequence similarity with the distal region of the N-domain of *E. coli* and *P. aeruginosa* IF2 (sub-domain N1) (Supplementary Fig. 1a, b). Correspondingly, the solution structure of the distal region of the *E. coli* IF2 N-domain (sub-domain N1) was shown to be highly similar to that of the N/G1 domain of ribosome-free *T. thermophilus* IF2[22].

It is interesting that the sequence encompassing the N1 sub-domain shares significant similarity with the region of the N-domain proximal to the G-domain (sub-domain N2), suggesting that sub-domain N2 may fold into a similar globular structure with the two repeats N1 and N2 connected by a linker forming the entire N-domain of IF2 (Supplementary Fig. 1a, c)[22,35]. The high sequence similarity between the N-domain of IF2 in *E. coli* and *P. aeruginosa* suggests a similar sub-domain arrangement

(Supplementary Fig. 1b, c). In the 70S-IC, the additional interactions provided by the N2 sub-domain further anchor IF2 on the ribosome, explaining how this region of IF2 stabilizes the binding of fMet-tRNA$_i^{fMet}$ to the P site of the 30S-IC and promotes joining of the 30 S and 50 S subunits[22,34,36]. Local refinement of this region yielded a 2.8-Å resolution map of the N2-globular sub-domain of IF2 interacting with helix h16 of 16 S rRNA (Fig. 7b; Supplementary Figs. 3 and 5d). An initial model of the N2 sub-domain was generated with AlphaFold2[37], and then fit and refined into the density. Using this approach, we traced 85 additional residues (248 to 332), of which residues 266 to 312 form the three-helix bundle sub-domain N2 in IF2 not previously included in the *E. coli* 70S-IC structures[10,21]. In the rotated 70S-IC (II-A), the buried surface area by extended IF2-GDPCP is ~5146 Å², of which ~1173 Å² is attributed to the N2 sub-domain.

While structure alignment of the three-helix bundle N2 sub-domain of ribosome-bound *P. aeruginosa* IF2 (residues 266 to 312) with the NMR structure of the *E. coli* sub-domain N1 (residues 2–50)[38] reveals reasonable homology with a root-mean-square-deviation (RMSD) value of 4.9 Å, alignment with the N/G1 domain from the crystal structure of ribosome-free *T. thermophilus* IF2[19,39] shows an almost perfect fit with RMSD of 1 Å (Supplementary Fig. 1d, e). The overall shape of the N2-globular sub-domain complements that of helix h16 in the shoulder region of the 30 S subunit, with residues Ser[277], Val[278], Lys[279] and Arg[259] forming the main interface and the extension that emerges from the N2 sub-domain wrapping around the minor groove of h16 (Fig. 7c, d). In the non-rotated 70S-IC the N2 region of compact IF2-GDP is not visible (Fig. 1a), indicating that the N2 sub-domain disengages from the 30 S shoulder and

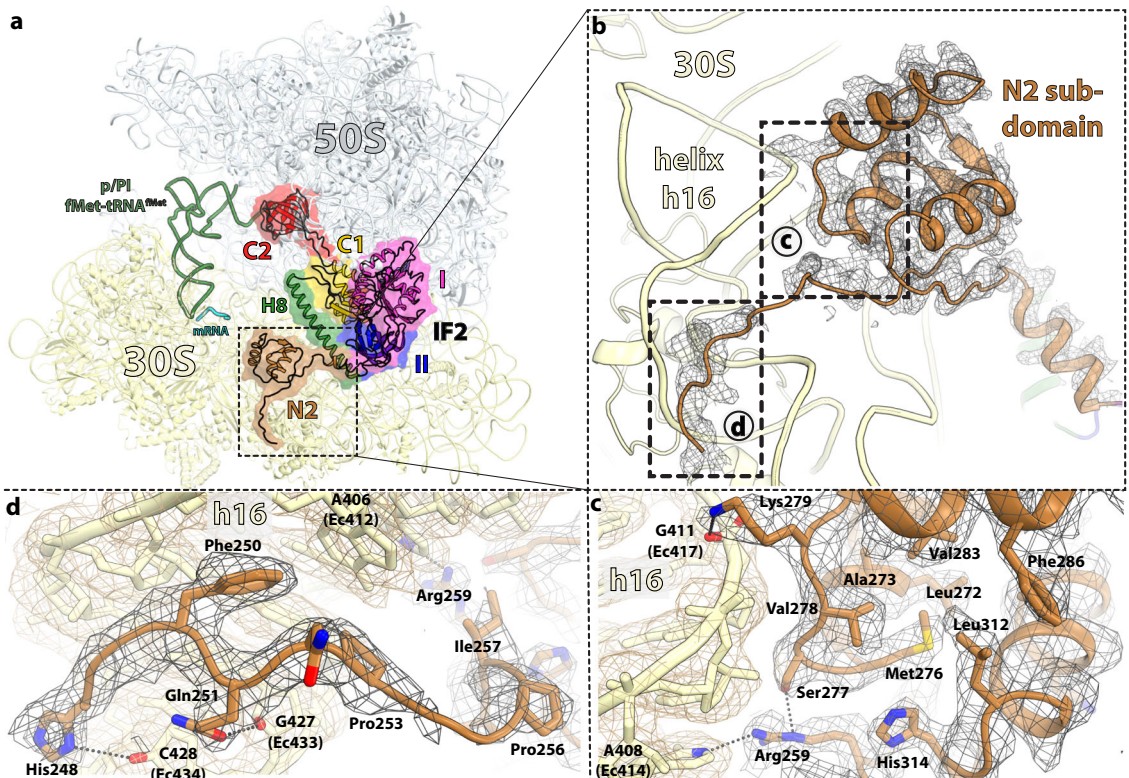

**Fig. 7 Structure of the N2 sub-domain of IF2 in the 70S-IC. a** Overview of extended IF2 in the 70S-IC with domains colored as in Fig. 1. The N2 sub-domain of IF2 (brown) interacts with the shoulder domain of the 30 S subunit. **b** Magnified view of the N2 sub-domain bound to the shoulder region of the 30 S subunit. The locally refined EM density map is shown as gray mesh. **c, d** Insets from panel **b**. **c** Interactions and surface complementarity between the N2-globular sub-domain of extended IF2 and helix h16 of 16 S rRNA. **d** The tail emerging from the N2 sub-domain of IF2 wraps around the minor groove of h16.

becomes labile upon back-rotation of the small subunit in the 70S-IC.

## Discussion

The compact conformation of IF2-GDP bound to the ribosome presented here suggests that IF2 undergoes large conformational changes during translation initiation. Structural and kinetic studies reported that IF2-GDP may remain bound to the 70S-IC[9,15,32,40], and our results show that IF2-GDP is primarily associated with the 70S-IC in the non-rotated state. Binding of translational GTPase factors to the ribosome is mutually exclusive and the persistence of ribosome-bound IF2-GDP at the end of initiation may account for the delay in binding of EF-Tu-GTP-aminoacyl-tRNA ternary complex to the ribosome[16,40]. It has been suggested that binding of the EF-Tu ternary complex may be facilitated by a "ready-to-leave" conformation of IF2-GDP that progressively loses interactions with the ribosome, possibly shifting the reaction equilibrium from the IF2-GDP-70S-IC to the EF-Tu-bound 70 S ribosome elongation complex[9]. Such state of IF2-GDP may be represented by our structure I-B, in which IF2 has lost interactions with the 30 S subunit and the SRL in the 50 S subunit.

Our preparation of the 70S-IC complex for cryo-EM structure determination deviates from the native initiation pathway, in which the 30 S subunit first forms a complex with mRNA, initiation factors IF1, IF2, IF3 and fMet-tRNA$_i^{fMet}$, followed by joining of the 50 S subunit and GTP hydrolysis by IF2[10]. Here we used purified 70 S ribosomes and added mRNA, fMet-tRNA$_i^{fMet}$, and IF2 in the presence of an excess of the GDPCP nucleotide, as previously reported (see Methods)[2,9,21]. Whether the IF2 conformations we observe on the 70S-IC represent on-pathway

intermediates is a legitimate concern. Two arguments suggest that the major structures (I-A/B & II-A/B) may nevertheless represent on-pathway, rather than off-pathway intermediates: i) the fact that the dissociation/association of IF2-GDP from/to the ribosome is reversible[9,12,15,32,40], and ii) the similarity of ribosomes II-A/II-B harboring extended IF2-GDPCP with the previous IF2–70S-IC structure obtained under native conditions (Supplementary Fig. 4)[10].

In the previous low-resolution cryo-EM structure of the *T. thermophilus* 70S-IC, IF2-GDP was observed in the extended conformation, which may seem at odd with the compact conformation of IF2-GDP reported here. A possible explanation could be attributed to the presence or absence of $P_i$ in IF2-GDP. However, in our structures, it is clear that GDP is bound without $P_i$. Similarly, the previous *T. th* IF2-GDP-70S-IC was prepared using GDP[9], and it is therefore unlikely that unreleased $P_i$ is bound to IF2. Another possibility considers the ribosome as a Brownian nanomachine, exploiting thermal motions for many of the conformational changes during protein synthesis[41]. It is conceivable that thermal motions of the ribosome could play an active role in promoting conformational changes of IF2. Both the *T. th* IF2-GDP-70S-IC and our *P. ae* 70S-IC were prepared by incubating the complexes at 37 °C (see ref. [9]. and Methods), which corresponds to the temperature at which *P. aeruginosa* ribosomes operate but is substantially lower than the optimum temperature for *T. thermophilus* (~75 °C). Therefore, a reduced flexibility at low temperature of the thermophilic ribosomes could have prevented IF2 from assuming the compact configuration.

In the crystal structures of ribosome-free GDP-bound and apo IF2, domain C1 occupies different positions relative to domains I-II, and domain C2 is not resolved likely due to intrinsic flexibility

(Supplementary Fig. 13a)[19]. The sw2-helix in ribosome-free IF2-GDP is disordered probably because of the absence of interaction with domain C1. In the 70S-IC, the space restraints imposed by the uL11-stalk may facilitate the re-location of domain C1 in IF2-GDP such that it docks against the sw2-helix in the G-domain. The flexibility of helix H8 in IF2 allows it to bend and accommodate the new position of domain C1 under the uL11-stalk, a feature that was not observed in the ribosome-free crystal structures of IF2 (Supplementary Fig. 13b).

The flexibility of the domains in IF2 made it difficult to rationalize how the signal of GTP hydrolysis in the G-domain is propagated to the distal domain C2 leading to accommodation of initiator tRNA into the P site[19,22]. Nevertheless, hydrolysis of GTP by IF2 is essential for separation of domain C2 from initiator tRNA[12] and for dissociation of the factor from the ribosome[11–13,15]. Based on our structures, we propose the following model for IF2-mediated translation initiation regarding the final steps of the process until IF2 is released. Upon GTP hydrolysis and $P_i$ release, the sw2-helix in the G-domain rotates and sw1 unfolds, releasing restraints on domain C1 and increasing its intrinsic flexibility. The unrestrained domain C1 loses interactions with the 30 S subunit, allowing the small ribosomal subunit to undergo reverse rotation. As the small subunit rotates back, the C1-domain swings around helix H8 of IF2 and relocates under the uL11-stalk of the 50 S subunit, forming a new interface with the G-domain bound to GDP. The large displacement and rotation of domain C1 pulls domain C2 away from fMet-tRNA$_i^{fMet}$, releasing initiator tRNA from IF2 into the P site and committing the ribosome to elongation (Supplementary Movie 1). In the rotated 70S-IC, the contact area between the ribosome and IF2-GDPCP is ~5146 Å$^2$. The transition of IF2 to the compact conformation combined with the reverse rotation of the ribosome drastically reduce the buried surface area by IF2-GDP in structures I-A (~2616 Å$^2$) and I-B (~2409 Å$^2$), providing the basis by which IF2-GDP eventually dissociates from the 70S-IC. To ensure we detected all conformational changes in IF2 and the ribosome that are present in the cryo-EM dataset reported here, we performed global 3D variability analysis using all ribosome particles, in both the rotated and non-rotated states. This analysis shows the same conformational changes in IF2 and the 70S-IC we describe using focused classification and refinement (see Supplementary Discussion, Supplementary Fig. 15, and Supplementary Movie 2).

In summary, cryo-EM of the bacterial 70S-IC revealed the elusive compact state of IF2-GDP bound to the ribosome. The structures offer insights into how remodeling of the switch regions, which is dependent on the nucleotide-bound state of the G-domain, is amplified into a cascade of domain rearrangements that promote irreversible release of fMet-tRNA$_i^{fMet}$ from IF2 and subsequent dissociation of IF2 from the ribosome. The model presented here illustrates how IF2 and the ribosome work hand-in-hand to guide the 70S-IC to an elongation competent complex.

## Methods

**Preparation of 70 S ribosomes, IF2, and initiator tRNA**. The full-length IF2 sequence was PCR amplified from *Pseudomonas aeruginosa* PAO1 genomic DNA and cloned into the pET28a-SMT3 plasmid (derived from pET28a, Novagen) containing an N-terminal 6x His-SMT3 tag. *Escherichia coli* BL21 (DE3) Star (Invitrogen) cells transformed with this construct were grown in the LB medium supplemented with 50 μg/mL kanamycin (Gold Biotechnology) to an absorbance of 0.8 at 600 nm before inducing expression of His-SMT3-*Pae*IF2 with 0.5 mM isopropylthiol-β-D-galactoside (IPTG, Gold Biotechnology) for 4 h at 37 °C. The cells were collected by centrifugation, flash frozen in liquid nitrogen and stored at –80 °C. To purify IF2, frozen cells were resuspended in lysis buffer at 4 °C (50 mM Tris pH 7.5, 500 mM NaCl, 20 mM imidazole, 1 mM PMSF, 1 mM β-mercaptoethanol) with one EDTA-free protease inhibitor tablet (Pierce, Thermo Scientific). The resuspended cells were lysed by passing several times through an LM20 high-pressure homogenizer (Microfluidics) operated at 15,000 psi. Cell debris were removed by centrifugation at 16,000 rpm (29,000 × g) at 4 °C for 45 min and filtered through a 0.22-μm filter (Millipore). The lysate was then loaded onto a 5 ml HisTrap HP (GE Healthcare) column and washed with a high salt buffer (lysis

buffer with 2 M NaCl) to remove bound nucleic acids and equilibrated back into lysis buffer. IF2 was eluted with a linear gradient of imidazole to 500 mM. Fractions containing His-SMT3-IF2 were pooled and incubated with the Ulp1 protease overnight at 4 °C to remove the SMT3-tag. The proteins were dialyzed into the lysis buffer and IF2 was separated from the SMT3-tag by passing through a reverse HisTrap column. IF2 was further purified by anion exchange on source Q (HR 16/10) and Superdex 200 16/60 (GE Healthcare) chromatography columns. Pure IF2 was concentrated to ~35 mg/ml, flash-frozen and stored at –80 °C.

The presence of nucleotide in purified IF2 was analyzed using a reverse-phase HPLC assay as described previously with minor modifications[42]. Briefly, 20 nmoles of purified IF2 or 3 nmoles of GTP or GDP (Millipore Sigma) in 70 μl of buffer (20 mM HEPES-KOH pH 7.5, 140 mM KCl) was treated with 0.6 μl of 70% perchloric acid at 22 °C for 2 min to denature the protein. Then, the pH was adjusted by adding 4 μl of 3 M Na-acetate pH 5.1 and the mixture centrifuged to pellet the precipitated protein. The supernatant was diluted 1:1 with HPLC column buffer A (100 mM K-phosphate pH 6.5, 10 mM tetrabutylammonium hydrogen sulphate) to 150 μl final volume. This sample was injected onto a PROTO 300 C4 HPLC column (10 × 250 mm) (Higgins Analytical) equilibrated in buffer A and eluted with a gradient of 0–50% buffer B containing 30% acetonitrile over 5 column volumes (Supplementary Fig. 2).

*E. coli* fMet-tRNA$_i^{fMet}$ was purified using established procedures[43]. Briefly, initiator tRNA$_i^{fMet}$ was expressed overnight in *E. coli* HB101 under the constitutive *lpp* promoter, extracted from the cells using phenol-chloroform and precipitated in 100% ethanol. tRNA$_i^{fMet}$ was purified by anion exchange source Q (HR 16/10) and reversed-phase chromatography PROTO 300 C4 HPLC (10 × 250 mm) (Higgins Analytical) columns. The deacylated tRNA$_i^{fMet}$ was then aminoacylated with methionyl-tRNA synthetase, formylated with methionyl-tRNA formyltransferase, and finally purified by reversed-phase HPLC on the C4 column. The 24-mer mRNA, containing a Shine-Dalgarno sequence and the initiation codon in the P site, with the sequence 5' GGC AAG GAG GUA AAA **AUG** UUC UAA 3' (start codon in bold), was chemically synthesized by Integrated DNA Technologies (Coralville, IA).

70 S ribosomes were purified from *P. aeruginosa* PAO1 as previously reported[44] with some modifications. Briefly, the cells were washed and lysed in buffer A (50 mM Tris-HCl pH 7.6, 10 mM MgCl$_2$, 100 mM NH$_4$Cl, 0.5 mM EDTA and 6 mM β-mercaptoethanol) using an Emulsiflex-C3 homogenizer (Avestin Inc.). The lysate was cleared by centrifugation at 18,000 rpm (36,000 × g) for 30 min at 4 °C and filter-sterilized through a 0.45 μm filter and stored at –80 °C. To isolate ribosomes, the lysate was layered on a 1.1 M sucrose cushion buffer (10 mM Hepes pH 7.6, 100 mM NH$_4$Cl, 14.5 mM MgCl$_2$, 0.5 mM EDTA, 6 mM β-mercaptoethanol) and spun at 43,000 rpm (214,000 × g) in a Type 45Ti rotor (Beckman) for 20 h at 4 °C. Ribosome pellets were resuspended in 10 mM Hepes pH 7.6, 100 mM NH$_4$Cl, 14.5 mM MgCl$_2$, 0.5 mM EDTA, 6 mM β-mercaptoethanol. Ribosomes were then purified through 10–50% sucrose density gradients in a SW32 rotor (Beckman) at 18,000 rpm (55,000 × g) at 4 °C for 19 h. The fractions containing 70 S ribosomes were collected, diluted to adjust Mg$^{2+}$ concentration to 10 mM and concentrated by centrifugation at 43,000 rpm (214,000 × g) at 4 °C. Pure 70 S ribosomes were resuspended and brought to the final buffer 10 mM Hepes pH 7.6, 60 mM NH$_4$Cl, 15 mM KCl, 10 mM MgCl$_2$, flash-frozen in liquid nitrogen and stored at –80 °C.

**Sample preparation, cryo-electron microscopy and data acquisition**. Translation initiation complex was programmed using 2 μM *P. aeruginosa* 70 S ribosome, 25 μM mRNA, 10 μM fMet-tRNA$_i^{fMet}$ and incubated in final 70 S buffer (10 mM Hepes 7.6, 60 mM NH$_4$Cl, 15 mM KCl, 10 mM MgCl$_2$, 1 mM β-mercaptoethanol) at 37 °C for 15 min. The ternary complex was then incubated with 30 μM IF2 and 2 mM GDPCP (Millipore Sigma) at room temperature for 15 min. The mixture containing 2 μM *P. aeruginosa* 70S-IC (4 μl) was applied to Quantifoil R2/1 gold 200 mesh grids (Electron Microscopy Sciences) which were pre-cleaned in a Solarus 950 plasma cleaner (Gatan). The grids were plunged-frozen in liquid nitrogen-cooled ethane using a Leica EM GP2 cryo-plunger. Grids were transferred into a Titan Krios G3i electron microscope (ThermoFisher Scientific) operating at 300 keV and equipped with a GIF Quantum LS energy filter (Gatan) and a K3 direct electron detector camera (Gatan). The image stacks (movies) were acquired in super-resolution mode with pixel size of 0.425 Å/pixel. Data collection was done in the EPU software (ThermoFisher Scientific) setup to record movies with 31 fractions with a total accumulated dose of ~31 e⁻/Å$^2$/movie. A total of 8056 image stacks were collected with a defocus ranging between –0.2 and –2.1 μm. The statistics of data acquisition are summarized in Supplementary Table 1.

**Cryo-EM data processing**. Data processing was done in cryoSPARC 3.1.0[45]. The image stacks were imported into cryoSPARC and gain-corrected. Image frames (fractions) were motion-corrected with dose-weighting and binned twice by Fourier cropping to the final pixel size of 0.85 Å/pixel. Patch contrast transfer function (CTF) estimation was performed on the motion-corrected micrographs. Based on relative ice thickness, CTF fit, and length and curvature of motion trajectories, 7259 micrographs were selected for further processing (Supplementary Fig. 3).

1,022,603 particles were picked using the circular "blob" picker in cryoSPARC and were filtered based on defocus adjusted power and pick scores to 878,576 particles.

Particles were subjected to two rounds of reference-free two-dimensional (2D) classification. After discarding bad particles, 494,268 particles were selected from 2D classification and used to generate ab-initio volumes with five groups. Two of the five groups containing 417,757 particles generated maps resembling the 70 S ribosome. The remaining three groups did not yield anything meaningful and were considered to contain not specimen-related particles which were discarded from further processing. Using 'heterogeneous refinement' in cryoSPARC these particles were further classified into three ribosome class averages. 180,951 particles were non-rotated 70 S (class average I) and 219,664 particles were rotated 70 S (class average II), with both class averages containing IF2 and P-site fMet-tRNA$_i$$^{fMet}$. The third class average comprising 17,142 particles gave a low-resolution reconstruction of the 70 S ribosome without IF2 and was therefore discarded. Focused 3D variability analysis performed on class average I with a mask around IF2 and the uL11-stalk showed large variability in the conformation of IF2 spanning both extended and compact conformation. Further classification of the non-rotated 70 S class average I particles based on variability focused in this region allowed to remove 32,939 particles with extended IF2, yielding 148,013 particles containing compact IF2. The best density for compact IF2 was observed in two sets of particles (I-A and I-B) that differ in the position of compact IF2 relative to 70 S and the degree of 30 S rotation. Similar focused 3D variability analysis around extended IF2 and the uL11-stalk on rotated 70 S ribosomes in class average II showed conformational variability of the uL11-stalk, sampling the outward and inward position, and the particles were accordingly classified into structures II-A (outward) and II-B (inward). Particles from structures I-A, I-B, II-A, and II-B were re-extracted to full-size (512 × 512 pixel box) and non-uniform and CTF refinement in cryoSPARC yielded reconstructions with overall resolutions ranging between 2.6 – 2.9 Å (Supplementary Fig. 5a, b).

For structures I-A and I-B, local refinement with signal subtraction using a soft mask around IF2 was performed, improving the overall resolution of compact IF2 to 3.4 Å (I-A) and 3.6 Å (I-B) (Supplementary Fig. 5c). Several regions of compact IF2 in structure I-A extend to a resolution of ~3.0 Å (Supplementary Fig. 5c). To improve the local resolution of the N2 sub-domain in the extended IF2, local refinement was performed using a mask around the N2 sub-domain and neighboring ribosomal elements of the 30 S shoulder. The best resolution map at 2.8 Å was obtained from local refinement of particles from both class averages I and II (Supplementary Figs. 3 and 5d). To optimally visualize features of different regions with varying local resolution, maps were locally sharpened using PHENIX.auto_sharpen in PHENIX 1.19.2 (Supplementary Table 1)[46].

**Model building and refinement**. As a starting model, the 30 S and 50 S subunits were taken from the structure of the *P. aeruginosa* 70 S ribosome [PDB 6SPG[44]] and rigid-body docked into the 2.7-Å cryo-EM map of structure II-A using UCSF Chimera 1.14[47]. The fMet-tRNA$_i$$^{fMet}$ in the p/PI state was taken from 3JCJ[21], rigid-body fit into the EM density, and adjusted in Coot[48]. The uL11-stalk region was adjusted and modeled into the density. The sequence of several ribosomal proteins and ribosomal RNA was revised according to the wild-type *P. aeruginosa* strain PAO1 and these regions were correspondingly remodeled. Model-to-map fit of each nucleotide in the 16 S, 23 S, and 5 S rRNAs was inspected visually and showed that several purines were modeled in the wrong *syn* conformation and were therefore remodeled into the *anti* conformation for a better fit into the EM density map. The high-resolution and quality of the EM map of structure II-A allowed building several Mg$^{2+}$ ions and water molecules, as well as the Zn$^{2+}$ ions in ribosomal proteins bL31 and bL36. Methylated nucleotides were modeled in the 16 S and 23 S rRNAs based on the *E. coli* high-resolution cryo-EM 70 S structure[49] and the corresponding conservation of RNA methyltransferases between *E. coli* and *P. aeruginosa* (Supplementary Table 2; Supplementary Fig. 14). Model of the *P.ae* 70 S ribosome was then rigid-body fit into the maps of structures I-A, I-B, and II-B by rotating the 30 S subunit by variable degrees and adjusting the uL11-stalk region into the density. In structures I-A and I-B, the initiator tRNA was rigid-body fit and adjusted in the p/P state.

For structures II-A and II-B, the initial model of extended *P. aeruginosa* IF2 was generated by homology modeling based on the ribosome-bound *E. coli* IF2 [PDB 3JCJ[21]] using MODELLER[50], rigid-body fit into the map, and then real-space refined in Coot. Complete models of 70 S, p/PI-tRNA, and extended IF2 with ordered solvent were real-space refined into EM maps of II-A and II-B for five cycles in PHENIX 1.19.2[46] including global energy minimization and group ADP refinement strategies along with base pair restraints for rRNA and initiator tRNA, together with Ramachandran and secondary structure restraints. The initial model for the N2 sub-domain residues 248–332 was generated using AlphaFold2[37], fit into the EM map and real-space refined in Coot and PHENIX. The model of the N2 sub-domain has real space correlation coefficient (RSCC) of 0.84 (Supplementary Fig. 5d; Supplementary Table 1), indicating excellent agreement with the density map[51].

The 70 S ribosome structures I-A and I-B were real-space refined for five cycles in PHENIX as described above. Separately, maps generated from a combination of signal subtraction and local refinement were used to model compact IF2 from structures I-A and I-B. Individual domains of IF2 were rigid body fit and manually adjusted into the density. Domain linkers and regions of visible discrepancies were manually built in Coot guided by the sequence and density. The complete model of compact IF2 containing residues 337–835 was real-space refined in PHENIX (with energy minimization, ADP refinement together with Ramachandran and secondary structure restraints) into the respective EM density maps obtained after focused refinement. The final models of compact IF2-GDP have RSCC of 0.82 (I-A) and

0.75 (I-B) (Supplementary Fig. 5c; Supplementary Table 1), indicating good agreement with the EM maps[51]. The lower RSCC for IF2-GDP in structure I-B is attributed to domain II which fits poorly into the EM density map (Supplementary Fig. 5c), consistent with the observation that IF2-GDP loses ribosome interactions in structure I-B upon the 30 S subunit back-rotation. A composite map combining the EM maps obtained after focused refinement of IF2 and the 70 S ribosome was generated for structures I-A and I-B, into which the complete model of 70S-IC bound to compact IF2-GDP was real-space refined in PHENIX. Resulting models were validated using the comprehensive validation tool for cryo-EM in PHENIX (Supplementary Table 1)[51].

The 30 S subunit rotation was calculated in PyMOL by aligning the 50 S subunit, excluding flexible regions (23 S rRNA residues 2090–2174 of the uL1-stalk, residues 1045–1099 of the uL11-stalk, and ribosomal proteins uL10 and uL11), of the four 70S-IC structures (I-A, I-B, II-A, II-B) with that of the non-rotated *P. aeruginosa* 70 S ribosome [PDB 6SPG[44]] and the angle measured between phosphate atoms of C1478 located in helix h44 of 16 S rRNA (considered the center of rotation) and U81 located in the foot region of the small subunit. The solvent accessible area between IF2 and the ribosome was calculated in PyMOL using the refined IF2 structures (I-A, I-B, II-A) with generated hydrogen atoms in the absence and presence of the ribosomal components within a radius of 30 Å around IF2. The numbering of helices in IF2 is according to Roll-Mecak et al.[52].

**Global 3D variability analysis of the 70S-IC cryo-EM dataset**. To visualize the conformational heterogeneity that is present within the 70S-IC in this cryo-EM dataset, we included all ribosome particles from class averages I and II and performed global 3D variability analysis in cryoSPARC 3.1.0[45]. The major variability component showed the 30 S subunit rotation relative to the 50 S subunit, coupled with IF2 domain rearrangements and fMet-tRNA$_i$$^{fMet}$ accommodation from the p/PI state to the p/P position. Particles were sorted into 20 clusters along this 3D conformational trajectory, and based on the extent of the 30 S rotation, clusters were ordered to reflect the transition from the rotated to the non-rotated 70S-IC. Clusters containing ribosomes with similar 30 S rotation were merged into distinct groups followed with non-uniform refinement[53], which yielded nine 70S-IC reconstructions with a global resolution ranging between 2.8 and 2.9 Å (Supplementary Fig. 15) (EM maps not provided). The reconstructions range from the ribosome with the most rotated 30 S subunit (~4.7°) containing extended IF2-GDPCP and fMet-tRNA$_i$$^{fMet}$ bound in the p/PI state, followed by structures that appear to represent intermediate transitions with one or more elements in IF2 and/or initiator tRNA with scattered density, and ending with a ribosome that is essentially non-rotated (~0.3°) bound to compact IF2-GDP and a well-ordered fMet-tRNA$_i$$^{fMet}$ accommodated into the P site. To visualize the conformational coupling that appears to exist between the ribosome, IF2 and initiator tRNA within the 70S-IC, the nine EM reconstructions were morphed together into a movie (Supplementary Movie 2).

**Figure and movie generation**. All figures showing atomic models were generated using PyMOL (The PyMOL Molecular Graphics System, Version 2.1.0 Schrödinger, LLC) or UCSF Chimera 1.14[47] and assembled with Adobe Illustrator (Adobe Inc.). Movies were made in UCSF ChimeraX 1.2[54].

**Reporting summary**. Further information on research design is available in the Nature Research Reporting Summary linked to this article.

## Data availability

The atomic coordinates were deposited in the RCSB Protein Data Bank (PDB) under accession codes 7UNQ (compact IF2-GDP, I-A), 7UNR (70S-IF2-GDP composite, I-A), 7UNT (compact IF2-GDP, I-B), 7UNU (70S-IF2-GDP composite, I-B), 7UNV (70S-IF2-GDPCP, II-A), 7UNW (70S-IF2-GDPCP, II-B), and 7UIU (IF2 N2 sub-domain). The cryo-EM maps have been deposited in the Electron Microscopy Data Bank (EMDB) under accession codes EMD-26628 (70 S, I-A), EMD-26629 (compact IF2-GDP from focused refinement, I-A), EMD-26630 (70S-IF2-GDP composite, I-A), EMD-26631 (70 S, I-B), EMD-26632 (compact IF2-GDP from focused refinement, I-B), EMD-26633 (70S-IF2-GDP composite, I-B), EMD-26634 [(70S-IF2-GDPCP, II-A), EMD-26635 (70S-IF2-GDPCP, II-B), and EMD-26553 (IF2 N2 sub-domain from focused refinement). The unaligned multi-frame cryo-EM micrographs have been deposited in the Electron Microscopy Public Image Archive (EMPIAR) with the accession code EMPIAR-11012.

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

## Acknowledgements

We thank members of the Gagnon laboratory for critical reading of the manuscript and suggestions. We are thankful to A. Miller for providing genomic DNA of the bacterium *P. aeruginosa*; J. Miller and P. Leiman for their assistance with culturing and cell lysis of *P. aeruginosa* PAO1; the Sealy Center for Structural Biology and Molecular Biophysics of The University of Texas Medical Branch at Galveston for providing critical infrastructure and expertise; K.-Y. Wong and J. Perkyns for computational support, N. Prokhorov for help and advice with the RELION 3 software package, and Y. Polikanov for sharing his custom-made Python scripts. This work was supported by NIH grant R01GM136936 (to M.G.G.), The Welch Foundation grant H-2032-20200401 (to M.G.G.), startup funds from The University of Texas Medical Branch (to M.G.G.), Rising Science and Technology Acquisition and Retention Program award from the University of Texas system (to M.G.G.), Pilot Grant from the Institute for Human Infections and Immunity at The University of Texas Medical Branch (to M.G.G.), and NIH grant P41GM103311 (to the

Resource for Biocomputing, Visualization, and Informatics at the University of California, San Francisco) for developing UCSF Chimera.

## Author contributions

R.S.B. and M.G.G. designed the project. R.S.B. performed experiments. R.S.B. and M.B.S. prepared the sample for cryo-EM. M.B.S. collected the cryo-EM data. R.S.B. and M.G.G. processed the cryo-EM data. R.S.B. and M.G.G. built the models and made the figures. R.S.B. and M.G.G. wrote the paper with contributions from M.B.S. All authors reviewed, edited, and approved the paper.

## Competing interests

The authors declare no competing interests.
