## [Peer Review File · Nature Communications]

Compact IF2 allows initiator tRNA accommodation into the P site and gates the ribosome to elongationREVIEWER COMMENTS

Reviewer #1 (Remarks to the Author):

Basu et al. report on a large-scale conformational switch of initiation factor 2 (IF2) that reveals how IF2 releases initiator tRNA on the bacterial ribosome at the final stage of translation initiation. The study is based on four cryo-EM structures of IF2 bound to the ribosome with fMet-tRNA^{fMet}. Two structures (II-A & II-B) depict IF2 with GDPCP in an elongated form bound to the rotated ribosome, similar as reported before, whereas two structures (I-A and I-B) show a novel compact form of IF2 with GDP bound to the non-rotated ribosome. Comparison of the ribosome-bound extended IF2 with GDPCP (II-A & II-B) vs. IF2-GDP in the compact state (I-A) reveals a cascade of conformational changes suggesting how GTP hydrolysis by IF2's G domain facilitates release of initiator tRNA from the distal C2 domain. Structure I-B shows further rearrangements of compact IF2-GDP that disrupt interactions with the ribosome providing insights into how GTP hydrolysis might promote dissociation of IF2. The work is overall technical sound and provides important novel information on the last step of translation initiation in bacteria. To warrant publication, the following points should be appropriately addressed.

Major points

1. The complex preparation deviates significantly from the native initiation pathway, as complexes were essentially prepared in the reverse direction: A mixture of IF2 with GDP and GDPCP was added in high excess to ribosomes pre-bound with initiator tRNA (most likely in the P/P state), i.e. to the final state of the initiation process. Therefore, it cannot be excluded that the observed structures represent off-pathway intermediates. Two arguments suggest that the major structures (I-A/B & II-A/B) may nevertheless represent on-pathway, rather than off-pathway intermediates: i) the fact that the dissociation/association of IF2-GDP from/to the ribosome is reversible and ii) the similarity of the present structures showing extended IF2-GDPCP with previous structures obtained under more native conditions (Kaledhonkar et al. Nature 2019). The present complex preparation vs. the native initiation process and its functional implications should be explicitly discussed in the main text.
2. The changes upon transition from the extended to the compact form of IF2 are not always clear in the Results section, largely because details on the extended state are missing. In particular, the authors should describe in more detail how both GTPase switch regions, sw1 & sw2, stabilize domain C1 in the extended form, as this will clarify how the loss of Pi facilitates the substantial rearrangement of the C1 domain. The interactions between C1 and C2 domains and the ribosome in the extended form should be described more clearly. Moreover, the authors should describe how the different conformations of the uL11 stalk in the presence of IF2-GDPCP compare with the uL11 stalk in the presence of IF2-GDP. Additional Supplementary Figure panels would be helpful to illustrate these points.
3. Figure 3: In Figure 3a, either ribosomal hallmarks in schematic form or an overview panel should be added to facilitate orientation. In Figure 3e,f similar, but distinct regions are shown for the compact and

extended forms of IF2, respectively. Supplementary Figure panels showing the two states in the same orientation or as superposition would help here to better understand the changes.

4. Structures of IF2 in the extended state have been reported before (e.g. Sprink et al. Science Adv. 2016 & Kaledhonkar et al. Nature 2019). The authors should show superpositions of the present with the most similar previous structures to clarify the similarities and potential differences, in particular with the 70S-IC structure from Kaledhonkar et al., which was obtained by a more native approach.

5. In the discussion, the authors describe a computational 3D variability analysis of their data revealing additional intermediates (IV-VII). However, the scattered densities for IF2 and/or initiator tRNA in these intermediates impede a functional interpretation. The discussion of this analysis should be moved to the Supplementary Information, as it provides no additional insights.

Minor Points

1. Lines 41-43: "Rotation of the switch 2 α -helix in the G-domain bound to GDP unlocks a cascade of large-domain movements in IF2 that propagate to the distal tRNA-binding domain C2". These changes can only be understood in relation to IF2's elongated form with GDPCP. The wording should be changed accordingly.

2. Line 128: "Compact IF2-GDP is bound to the 70S-IC transitioning to the non-rotated state". The term "transitioning" is misleading and should be removed here and below (lines 139, 266), as IF2-GDP is essentially bound to the non-rotated ribosome ($\sim 1^\circ$ rotation is neglectable).

3. The final concentration of 70S-ICs applied on cryo-EM grids is unclear and should be clarified in the Methods.

Reviewer #2 (Remarks to the Author):

The article by Basu and colleagues presents cryo-electron microscopy structures of complexes of *Pseudomonas aeruginosa* 70S ribosomes with IF2-GDP and initiator fMet-tRNA, corresponding to a final stage of translation initiation, just before the ribosome enters the elongation phase of translation. Previously, several kinetic and structural studies have focused on the events which, starting from a 30S initiation complex, lead to the formation of an elongation competent 70S complex. Collectively, the kinetic analyses have indicated the existence of a number of conformational changes involving the ribosome, IF2 and initiator tRNA and have yielded the kinetic parameters of the individual steps. On the other hand, the structural studies have revealed the molecular bases of some of the conformational changes occurring during the formation of a 70S EC.

Conformational changes of the G-domain switch 2 α -helix as a function of the guanosine nucleotide bound to IF2, movements of IF2 with respect to r-protein L11, separation of IF2 C2 from the CCA end of

initiator tRNA, repositioning of fMet-tRNA from the P/I to the P/P site, rotation and back rotation of the ribosomal subunits have been described in already published literature.

Thus, there is little new information in this article with respect to what is already known, aside from the “hitherto unseen compact state of IF2-GDP bound to the ribosome”. The present detailed high-resolution description of the cascade of large-domain movements in GDP-bound IF2 which eventually propagate to the distal tRNA-binding domain C2 and causes its relocation 35 angstroms away from the tRNA and its collapse on C1 is undoubtedly impressive. However, the existence of the compact structure of IF2 poses a number of problems which have not been addressed by the authors. In fact, in the cryoEM structure presented by Myasnikov et al. (NSMB, 12, 1145, 2005), IF2-GDP is seen in an elongated conformation and, compared to the position occupied in the initiation complex containing IF2-GMPPCP, the C2 domain has moved away from the acceptor end of the initiator tRNA which occupies the canonical P site while the IF2 G domain has moved with respect to the SRL and L11 and the extent of the IF2-ribosome interaction is diminished so that the IF2 position is indicated as being “ready to leave”.

By and large, this article describes the same or closely related events. Thus, a fundamental question is how is it possible that IF2-GDP can assume such different structures such as an extended and a compact conformation within 70S complexes having essentially the same composition. A possible explanation that the difference is due to the presence or absence of the phosphate must be ruled out because Basu et al. clearly state that the phosphate is not seen in their structure whereas in the case of Myasnikov et al. the complex was prepared using GDP so that it is unlikely that an unreleased phosphate is bound to IF2. Perhaps a way out could be found if one assumes that a Brownian motor mechanism and the flexibility of the ribosome play an active role in promoting the conformational changes of IF2. If this is true then it is possible to imagine that the different origin of the ribosomes (*T. thermophilus* vs *P. aeruginosa*) is responsible for the observed differences, insofar as the temperature at which the complexes were made corresponds to the temperature at which *P. aeruginosa* ribosomes operate but is substantially lower than the optimum temperature of *T. thermophilus*. Thus, a reduced flexibility at low temperature of the thermophilic ribosomes could have prevented IF2 from assuming the compact configuration.

Aside from these considerations, I would like to mention that Basu et al have not made any attempt to discuss the movements of IF2 with respect to L11 in light of the IF2-L11 FRET results published by Qin et al. (Biochemistry, 48, 4699, 2009).

Finally, I would have expected a discussion concerning the issue of IF2 dissociation from the ribosome at the end of the initiation phase of translation. The results of this article, as well as those of Myasnikov et al. demonstrate that IF2 remains ribosome-bound even after the dissociation of the interaction between the C2 domain and the acceptor end of fMet-tRNA and after the substantial reduction of its surface interacting with the ribosome which results from its conformational changes. That IF2 does not dissociate from the ribosome at the end of the initiation phase is also indicated by the persistence of the IF2-L11 FRET signal (Qin et al. Biochemistry, 48, 4699, 2009). Thus, it seems likely that after the adjustment of the initiator tRNA in the P site, IF2 does not fall off but moves away from its original binding site clearing the way for the binding of the EF-Tu-GTP-aa-tRNA ternary complex (Tomsic et al. EMBO J 19, 2127, 2000) and that its dissociation occurs only upon the binding of EF-G.

Reviewer #3 (Remarks to the Author):

The manuscript by Basu et al. addresses the structural transitions of bacterial translation initiation factor IF2 on the ribosome during the final stages of translation initiation. For this, the authors assembled a 70S initiation complex from the bacterium *P. aeruginosa* in the presence of IF2, initiator tRNA, mRNA and a mix of GDP and the non-hydrolysable analogue GDPCP to obtain a mix of GTP and GDP states in the sample. Cryo-EM, image processing and structure sorting allowed to separate two main states whose structures were refined with focused classifications and refinements. The cryo-EM analysis appears sound technically and the structures are resolved to better than 3 Å resolution. The results are interesting as a compact conformation of 70S-bound IF2 is found in which the C-terminal part is retracted away from the P-site tRNA and hence allowing it to accommodate into a state that is primed for the elongation phase of protein synthesis. The compact conformation of IF2 implies a number of conformational rearrangements of the IF2 domains. It is also interesting to see that the N-terminal region of IF2, which is known to promote 50S subunit joining onto the 30S IC, interacts with the 30S in the GTP-analogue state but dissociates in the GDP state along with a conformational back-rotation of the 30S subunit. A key question that could probably be addressed from these structures is the role of Pi release (not discussed yet). To me, the structural and conformational transition occurs not upon GTP hydrolysis but upon Pi release, hence when IF2-bound GDP-Pi transits to IF2/GDP. Probably this is the meaning of the observations made here and would be worth clarifying / discussing. In addition to the main findings, there are also interesting observations on chemical modifications (in the figures, but not discussed). Altogether, the paper is concise and clear and presents interesting results.

Detailed points:

- IF2 contains a N-terminal of highly variable length depending on the organism. It would be worth including a sequence alignment to illustrate this. Apparently, in *P. aeruginosa* it is rather long and over 200 residues are not visible in the structure (to be mentioned/explained)
- abstract: “distinct”: from what? Maybe better: precise (same issue in the introduction); “after subunit joining” is misleading, in fact it is upon subunit joining that hydrolysis occurs (see kinetics studies), is the first and fastest event before conformational changes occur (some of which are depending on the Pi release). “Releasing tRNA... into the P-site” is also not clear, it is actually already there but needs to accommodate (that term would be better and is used later). Same for “hands tRNA to the P-site”.
- introduction: “large 50S subunit” should better say large subunit (50S)
- lines 83 & 128: “transitions” doesn’t tell what is happening, in fact it back rotates
- line 90: “hands tRNA into P-site” is incorrect concept (see comments on abstract), better would be accommodates

- lines 108/109: those two states appear similar to those described in the Myasnikov et al. NSMB 2005 paper (suggested to be GTP and GDP states at the time), is that the case? Maybe worth discussing there
- as the structure sorting and refinements heavily rely on focused classifications and refinements it may be worth mentioning some key references such as Curr Op Struct Biol 2017, eLife 2018, Biosci Rep 2018 etc.
- line 129: if IF2 is resolved it is not clear why additional focused refinement was needed, maybe reformulate
- focused refinement with signal subtraction: did this include re-centring? that could improve image processing even further
- 155: what is the issue with the poorly defined linker, does it affect the conformation or the conclusions?
- line 165: sw2 alpha helix: this is rather buried next to the active site in the G domain, what does the rotation imply for neighbouring secondary structure elements and side-chains of IF2 (once it has left the SRL region)?
- line 168: "GTP hydrolysis must occur" and also Pi release. Mechanistically this is an important step
- buried surface: the change is about 10%, which not sound that large/significant. Maybe better to focus on the loss of specific interactions with the helix 6 region etc.
- line 239 and following paragraph: the idea that the N-domain holds on the 30S and promotes subunit joining with the 50S was originally described in reference 18 (could be mentioned from the beginning). The fact that additional interactions provided by the N-domain anchor IF2 on the ribosome is consistent with observations made in ref. 18. Interesting that the interaction is released upon back-rotation of the 30S subunit
- line 282: how does the crystal structure of IF2 (Eiler et al, at the time suggested to be in a different conformation than on the ribosome) now compare with the IF2 conformation seen in the compact state? Is shown in Suppl but not discussed
- line 288: "we propose the following model for IF2-mediated translation initiation": maybe add: regarding the final steps of the process until IF2 is released (considering that IF2 and IF1/3 have a role in pre-initiation and early 30SIC formation etc.)
- 291/292: conformational changes of IF2 and subunit rotation appear to be induced by Pi release; the question is what starts the process: GTP hydrolysis is required but not sufficient, it needs Pi release (see published kinetics studies)
- angle estimation of 30S rotation: around 1° it may become difficult to have precise values. For the calculation, were the 50S subunit parts aligned to each other and then the angle measured on the 30S? (23 S rRNA is indicated, full 50S might be better, without the flexible parts). This gives more precise values and avoids getting averaged value effects with the 50S subunit not being well aligned
- kink in the tip of helix 8: any particular amino acid sequence that would favour this (Gly etc.)?

- “focused EM density” sounds imprecise, maybe better: cryo-EM density obtained after focused refinement (text and legends)
- “classes” should be better called class averages
- clashes in the atomic model should be checked for / refined (see validation report)
- cryo-EM maps, half maps, masks etc. and fully refined atomic models should be deposited in the EMDB and PDB, and representative data deposited onto the EMPIAR data base

Thank you for the opportunity to review this work. Bruno Klaholz

We would like to extend our gratitude to all three reviewers for their constructive comments and suggestions on our manuscript. All the suggestions and changes have been incorporated in the revised version of the manuscript. The new and/or edited text is blue.

REVIEWER COMMENTS

Reviewer #1 (Remarks to the Author):

Basu et al. report on a large-scale conformational switch of initiation factor 2 (IF2) that reveals how IF2 releases initiator tRNA on the bacterial ribosome at the final stage of translation initiation. The study is based on four cryo-EM structures of IF2 bound to the ribosome with fMet-tRNA^{fMet}. Two structures (II-A & II-B) depict IF2 with GDPCP in an elongated form bound to the rotated ribosome, similar as reported before, whereas two structures (I-A and I-B) show a novel compact form of IF2 with GDP bound to the non-rotated ribosome. Comparison of the ribosome-bound extended IF2 with GDPCP (II-A & II-B) vs. IF2-GDP in the compact state (I-A) reveals a cascade of conformational changes suggesting how GTP hydrolysis by IF2's G domain facilitates release of initiator tRNA from the distal C2 domain. Structure I-B shows further rearrangements of compact IF2-GDP that disrupt interactions with the ribosome providing insights into how GTP hydrolysis might promote dissociation of IF2. The work is overall technical sound and provides important novel information on the last step of translation initiation in bacteria. To warrant publication, the following points should be appropriately addressed.

Major points

1. The complex preparation deviates significantly from the native initiation pathway, as complexes were essentially prepared in the reverse direction: A mixture of IF2 with GDP and GDPCP was added in high excess to ribosomes pre-bound with initiator tRNA (most likely in the P/P state), i.e. to the final state of the initiation process. Therefore, it cannot be excluded that the observed structures represent off-pathway intermediates. Two arguments suggest that the major structures (I-A/B & II-A/B) may nevertheless represent on-pathway, rather than off-pathway intermediates: i) the fact that the dissociation/association of IF2-GDP from/to the ribosome is reversible and ii) the similarity of the present structures showing extended IF2-GDPCP with previous structures obtained under more native conditions (Kaledhonkar et al. Nature 2019). The present complex preparation vs. the native initiation process and its functional implications should be explicitly discussed in the main text.

Thank you for this suggestion. We have added the following to the Discussion section:

“Our preparation of the 70S-IC complex for cryo-EM structure determination deviates from the native initiation pathway, in which the 30S subunit is first incubated with mRNA, fMet-tRNA_i^{fMet}, initiation factors IF1, IF2 and IF3, and finally the 50S subunit is added to form the 70S-IC¹⁰. Here we used purified 70S ribosomes and added mRNA, fMet-tRNA_i^{fMet}, and IF2 in the presence of an excess of the GDPCP nucleotide, as previously reported (see Methods)^{2,9,20}.

Whether the IF2 conformations we observe on the 70S-IC represent on-pathway intermediates is a legitimate concern. Two arguments suggest that the major structures (I-A/B & II-A/B) may nevertheless represent on-pathway, rather than off-pathway intermediates: i) the fact that the dissociation/association of IF2-GDP from/to the ribosome is reversible^{9,12,15,31,39}, and ii) the similarity of ribosomes II-A/II-B harboring extended IF2-GDPCP with the previous IF2-70S-IC structure obtained under native conditions (Supplementary Fig. 4)¹⁰.”

2. The changes upon transition from the extended to the compact form of IF2 are not always clear in the Results section, largely because details on the extended state are missing. In particular, the authors should describe in more detail how both GTPase switch regions, sw1 & sw2, stabilize domain C1 in the extended form, as this will clarify how the loss of Pi facilitates the substantial rearrangement of the C1 domain.

To illustrate the interactions between the G- and C1-domains in extended IF2, we added Supplementary Fig. 10, which is composed of 5 panels. Panel c shows how sw2 interacts with domain C1 and two panels (d, e) show interactions between sw1 and the C1-domain. In panels c – e, the EM density is displayed as mesh.

We also added the following to the text:

“In the extended conformation of IF2-GDPCP, both switch regions contact domain C1 (Fig. 3f; Supplementary Fig. 10), contributing to its rigidity. Upon GTP hydrolysis and P_i release, sw1 becomes disordered and loses interactions with domain C1 while sw2 is remodeled. This releases structural restraints on domain C1 which is then free to relocate under the uL11-stalk in compact IF2 (Fig. 5a, e).”

The interactions between C1 and C2 domains and the ribosome in the extended form should be described more clearly.

The new Supplementary Fig. 11 addresses this point. Panel b shows the location of domain C1 within the cleft formed by ribosomal protein uL14, helix h5 of 16S rRNA, and protein uS12. Three panels (c – e) illustrate the interactions between domain C2 of extended IF2 and the ribosome. In panels a – e, the EM density is shown as mesh.

The new Supplementary Fig. 11 is labeled:

“Supplementary Fig. 11. Interactions between the C1- and C2-domains of extended IF2 and the ribosome in the 70S-IC.”

In the text we added the following:

“This is in contrast with the location of domain C2 in extended IF2-GDPCP on the rotated ribosome which, in addition to interacting with the CCA-end of fMet-tRNA_i^{fMet}, also contacts H89, H71 and ribosomal protein uL16 (Supplementary Fig. 11a, c–e).”

“In structures I-A and I-B with compact IF2-GDP, the uL11-stalk is in the inward position similar to that observed in structure II-B (Fig. 6a; Supplementary Fig. 6). This allows nucleotide A1057 (A1067) at the tip of H43 to form a stacking interaction with conserved Tyr⁷²³ in domain C1 that moved from the inter-subunit space in extended IF2 (Supplementary Fig. 11a–b) to its location under the uL11-stalk in compact IF2 (Fig. 6a, e; Supplementary Fig. 9).”

Moreover, the authors should describe how the different conformations of the uL11 stalk in the presence of IF2-GDPCP compare with the uL11 stalk in the presence of IF2-GDP. Additional Supplementary Figure panels would be helpful to illustrate these points.

The conformations of the uL11-stalk are now compared for each reported structure in Supplementary Fig. 6b.

In the text, it reads:

“In structures I-A and I-B with compact IF2-GDP, the uL11-stalk is in the inward position similar to that observed in structure II-B (Fig. 6a; Supplementary Fig. 6).”

3. Figure 3: In Figure 3a, either ribosomal hallmarks in schematic form or an overview panel should be added to facilitate orientation.

We now include an inset overview in Fig. 3a.

In Figure 3e,f similar, but distinct regions are shown for the compact and extended forms of IF2, respectively. Supplementary Figure panels showing the two states in the same orientation or as superposition would help here to better understand the changes.

Both panels e and f of figure 3 are now shown in the same orientation. Panel f also shows the proximity of domain C1 in extended IF2 to both switches 1 (sw1) and 2 (sw2). Detailed interactions are shown in the new Supplementary Fig. 10.

4. Structures of IF2 in the extended state have been reported before (e.g. Sprink *et al.* *Science Adv.* 2016 & Kaledhonkar *et al.* *Nature* 2019). The authors should show superpositions of the present with the most similar previous structures to clarify the similarities and potential differences, in particular with the 70S-IC structure from Kaledhonkar *et al.*, which was obtained by a more native approach.

We now show superposition of our IF2-GDPCP with that obtained by Kaledhonkar *et al.* *Nature* 2019 and Sprink *et al.* *Sci. Adv.* 2016 (see new Supplementary Fig. 4). The superimpositions yielded an RMSD value of 1.6Å.

5. In the discussion, the authors describe a computational 3D variability analysis of their data revealing additional intermediates (IV-VII). However, the scattered densities for IF2 and/or initiator tRNA in these intermediates impede a functional interpretation. The discussion of this

analysis should be moved to the Supplementary Information, as it provides no additional insights.

Thank you for this suggestion. We completely agree. The global 3D variability analysis discussion has been moved to Supplementary Information.

Minor Points

1. Lines 41-43: “Rotation of the switch 2 α -helix in the G-domain bound to GDP unlocks a cascade of large-domain movements in IF2 that propagate to the distal tRNA-binding domain C2”. These changes can only be understood in relation to IF2’s elongated form with GDPCP. The wording should be changed accordingly.

Excellent point. We changed the text to:

“Relative to GTP-bound IF2, rotation of the switch 2 α -helix in the G-domain bound to GDP unlocks a cascade of large-domain movements in IF2 that propagate to the distal tRNA-binding domain C2.”

2. Line 128: “Compact IF2-GDP is bound to the 70S-IC transitioning to the non-rotated state”. The term “transitioning” is misleading and should be removed here and below (lines 139, 266), as IF2-GDP is essentially bound to the non-rotated ribosome ($\sim 1^\circ$ rotation is neglectable).

Correct, thank you! All instances hinting to a “transition” have been removed.

3. The final concentration of 70S-ICs applied on cryo-EM grids is unclear and should be clarified in the Methods.

The concentration of 70S-IC applied to the grids is now explicitly mentioned in the Methods section. It now reads:

“The mixture containing 2 μ M *P. aeruginosa* 70S-IC (4 μ l) was applied to Quantifoil R2/1 gold 200 mesh grids (Electron Microscopy Sciences) which were pre-cleaned in a Solarus 950 plasma cleaner (Gatan).”

Reviewer #2 (Remarks to the Author):

The article by Basu and colleagues presents cryo-electron microscopy structures of complexes of *Pseudomonas aeruginosa* 70S ribosomes with IF2-GDP and initiator fMet-tRNA, corresponding to a final stage of translation initiation, just before the ribosome enters the elongation phase of translation. Previously, several kinetic and structural studies have focused on the events which, starting from a 30S initiation complex, lead to the formation of an elongation competent 70S complex. Collectively, the kinetic analyses have indicated the existence of a number of conformational changes involving the ribosome, IF2 and initiator tRNA and have yielded the kinetic parameters of the individual steps. On the other hand, the structural studies have revealed the molecular bases of some of the conformational changes occurring during the formation of a 70S EC.

Conformational changes of the G-domain switch 2 α -helix as a function of the guanosine nucleotide bound to IF2, movements of IF2 with respect to r-protein L11, separation of IF2 C2 from the CCA end of initiator tRNA, repositioning of fMet-tRNA from the P/I to the P/P site, rotation and back rotation of the ribosomal subunits have been described in already published literature. Thus, there is little new information in this article with respect to what is already known, aside from the “hitherto unseen compact state of IF2-GDP bound to the ribosome”.

Here we have to respectfully disagree with this assessment, and below is why:

In the latest structural studies of IF2 bound to the ribosome, IF2 was observed in a single conformation, the extended form. The crystal structures of IF2 in the apo and GDP-bound forms determined by Eiler D *et al.* PNAS 2013 offer very little insights regarding the conformation of the sw2 helix, simply because in the GDP form, the sw2 helix in IF2 was not visible in their electron density map. Although the crystal structure of ribosome-free IF2-GDP shows a compact form, the relevance of this conformation was unclear in the absence of a structure of ribosome-bound IF2-GDP. The only available structure of IF2-GDP bound to the 70S ribosome is the cryo-EM by Myasnikov *et al.* NSMB, 12, 1145, 2005. However, the low-resolution of this structure limits the conclusions that can be derived regarding the conformation of the sw2 helix and other secondary structure elements in IF2.

The separation of domain C2 of IF2 from tRNA was observed in the cryo-EM reconstruction by Myasnikov *et al.* NSMB, 12, 1145, 2005. However, the low-resolution of the structure did not allow to describe the mechanism driving the separation.

Yes, we agree that the repositioning of initiator tRNA from the P/I to the P/P state was previously observed (and also expected) as a consequence of the ribosome 30S subunit back rotation.

While movement of IF2 relative to ribosomal protein uL11 has been described based on FRET studies (Qin *et al.* Biochemistry, 2009) and Myasnikov *et al.* NSMB, 12, 1145, 2005 hinted to a movement of the G-domain relative to the uL11-stalk from their low-resolution cryo-EM of the *T. thermophilus* 70S-IC, no structural model could be built of IF2 in such state of ribosome binding.

Therefore, the structures presented here report many aspects and details that were completely missing from previous structural studies of ribosome-bound IF2.

The present detailed high-resolution description of the cascade of large-domain movements in GDP-bound IF2 which eventually propagate to the distal tRNA-binding domain C2 and causes its relocation 35 angstroms away from the tRNA and its collapse on C1 is undoubtedly impressive. However, the existence of the compact structure of IF2 poses a number of problems which have not been addressed by the authors. In fact, in the cryoEM structure presented by Myasnikov et al. (NSMB, 12, 1145, 2005), IF2-GDP is seen in an elongated conformation and, compared to the position occupied in the initiation complex containing IF2-GMPPCP, the C2 domain has moved away from the acceptor end of the initiator tRNA which occupies the canonical P site while the IF2 G domain has moved with respect to the SRL and L11 and the extent of the IF2-ribosome interaction is diminished so that the IF2 position is indicated as being “ready to leave”.

By and large, this article describes the same or closely related events. Thus, a fundamental question is how is it possible that IF2-GDP can assume such different structures such as an extended and a compact conformation within 70S complexes having essentially the same composition. A possible explanation that the difference is due to the presence or absence of the phosphate must be ruled out because Basu et al. clearly state that the phosphate is not seen in their structure whereas in the case of Myasnikov et al. the complex was prepared using GDP so that it is unlikely that an unreleased phosphate is bound to IF2. Perhaps a way out could be found if one assumes that a Brownian motor mechanism and the flexibility of the ribosome play an active role in promoting the conformational changes of IF2. If this is true then it is possible to imagine that the different origin of the ribosomes (*T. thermophilus* vs *P. aeruginosa*) is responsible for the observed differences, insofar as the temperature at which the complexes were made corresponds to the temperature at which *P. aeruginosa* ribosomes operate but is substantially lower than the optimum temperature of *T. thermophilus*. Thus, a reduced flexibility at low temperature of the thermophilic ribosomes could have prevented IF2 from assuming the compact configuration.

Thank you for this suggestion. We also believe it is important to clarify why we now observe the compact state of IF2-GDP while in the *T. thermophilus* complex only the extended form of IF2 was reported. We included the following in the Discussion section:

“In the previous low-resolution cryo-EM structure of the *T. thermophilus* 70S-IC, IF2-GDP was observed in the extended conformation, which may seem at odd with the compact conformation of IF2-GDP reported here. A possible explanation could be attributed to the presence or absence of P_i in IF2-GDP. However, in our structures, it is clear that GDP is bound without P_i . Similarly, the previous *T. th* IF2-GDP-70S-IC was prepared using GDP⁹, and it is therefore unlikely that unreleased P_i is bound to IF2. Another possibility considers the ribosome as a Brownian nanomachine, exploiting thermal motions for many of the conformational changes during protein

synthesis⁴⁰. It is conceivable that thermal motions of the ribosome could play an active role in promoting conformational changes of IF2. Both the *T. th* IF2-GDP-70S-IC and our *P. ae* 70S-IC were prepared by incubating the complexes at 37°C (see ref. ⁹ and Methods), which corresponds to the temperature at which *P. aeruginosa* ribosomes operate but is substantially lower than the optimum temperature for *T. thermophilus* (~75°C). Therefore, a reduced flexibility at low temperature of the thermophilic ribosomes could have prevented IF2 from assuming the compact configuration.”

Aside from these considerations, I would like to mention that Basu et al have not made any attempt to discuss the movements of IF2 with respect to L11 in light of the IF2-L11 FRET results published by Qin et al. (Biochemistry, 48, 4699, 2009).

Thank you for this excellent suggestion. We have incorporated the following:

“The close proximity of the N-domain of uL11 and the G-domain of IF2 in structure II-A is in agreement with previous FRET studies reporting that upon 70S-IC formation, the G-domain of IF2 moves closer to the uL11-NTD both in the presence of GTP or a non-hydrolysable GTP analogue³¹.”

Finally, I would have expected a discussion concerning the issue of IF2 dissociation from the ribosome at the end of the initiation phase of translation. The results of this article, as well as those of Myasnikov et al. demonstrate that IF2 remains ribosome-bound even after the dissociation of the interaction between the C2 domain and the acceptor end of fMet-tRNA and after the substantial reduction of its surface interacting with the ribosome which results from its conformational changes. That IF2 does not dissociate from the ribosome at the end of the initiation phase is also indicated by the persistence of the IF2-L11 FRET signal (Qin et al. Biochemistry, 48, 4699, 2009). Thus, it seems likely that after the adjustment of the initiator tRNA in the P site, IF2 does not fall off but moves away from its original binding site clearing the way for the binding of the EF-Tu-GTP-aa-tRNA ternary complex (Tomsic et al. EMBO J 19, 2127, 2000) and that its dissociation occurs only upon the binding of EF-G.

Thank you for this suggestion. We addressed this point by including the following in the Discussion section:

“Binding of translational GTPase factors to the ribosome is mutually exclusive and the persistence of ribosome-bound IF2-GDP at the end of initiation may account for the delay in binding of EF-Tu-GTP-aminoacyl-tRNA ternary complex to the ribosome^{16,39}. It has been suggested that binding of the EF-Tu ternary complex may be facilitated by a “ready-to-leave” conformation of IF2-GDP that progressively loses interactions with the ribosome, possibly shifting the reaction equilibrium from the IF2-GDP-70S-IC to the EF-Tu-bound 70S ribosome elongation complex⁹. Such state of IF2-GDP may be represented by our structure I-B, in which IF2 has lost interactions with the 30S subunit and the SRL in the 50S subunit.”

Reviewer #3 (Remarks to the Author):

The manuscript by Basu et al. addresses the structural transitions of bacterial translation initiation factor IF2 on the ribosome during the final stages of translation initiation. For this, the authors assembled a 70S initiation complex from the bacterium *P. aeruginosa* in the presence of IF2, initiator tRNA, mRNA and a mix of GDP and the non-hydrolysable analogue GDPCP to obtain a mix of GTP and GDP states in the sample. Cryo-EM, image processing and structure sorting allowed to separate two main states whose structures were refined with focused classifications and refinements. The cryo-EM analysis appears sound technically and the structures are resolved to better than 3 Å resolution. The results are interesting as a compact conformation of 70S-bound IF2 is found in which the C-terminal part is retracted away from the P-site tRNA and hence allowing it to accommodate into a state that is primed for the elongation phase of protein synthesis. The compact conformation of IF2 implies a number of conformational rearrangements of the IF2 domains. It is also interesting to see that the N-terminal region of IF2, which is known to promote 50S subunit joining onto the 30S IC, interacts with the 30S in the GTP-analogue state but dissociates in the GDP state along with a conformational back-rotation of the 30S subunit. A key question that could probably be addressed from these structures is the role of Pi release (not discussed yet). To me, the structural and conformational transition occurs not upon GTP hydrolysis but upon Pi release, hence when IF2-bound GDP-Pi transits to IF2/GDP. Probably this is the meaning of the observations made here and would be worth clarifying / discussing.

In addition to the main findings, there are also interesting observations on chemical modifications (in the figures, but not discussed). Altogether, the paper is concise and clear and presents interesting results.

Detailed points:

- IF2 contains a N-terminal of highly variable length depending on the organism. It would worth including a sequence alignment to illustrate this. Apparently, in *P. aeruginosa* it is rather long and over 200 residues are not visible in the structure (to be mentioned/explained)

To illustrate this point, we now provide a sequence alignment of the N-region of IF2 from *E. coli*, *P. aeruginosa*, and *T. thermophilus* in Supplementary Figure 1. To further clarify regions of the N-domain that were studied before, we call the distal region the N1 sub-domain and the region proximal to the G-domain, the N2 sub-domain. In Supplementary Fig. 1, we also provide superimpositions of the available structures of the N-domain regions of IF2 with our structure of the N2 sub-domain. The superposition RMSD values are provided and the text has been modified accordingly.

- abstract: “distinct”: from what? Maybe better: precise (same issue in the introduction); “after subunit joining” is misleading, in fact it is upon subunit joining that hydrolysis occurs (see kinetics studies), is the first and fastest event before conformational changes occur (some of

which are depending on the Pi release). “Releasing tRNA... into the P-site” is also not clear, it is actually already there but needs to accommodate (that term would be better and is used later). Same for “hands tRNA to the P-site”.

In the abstract, we changed the word “distinct” to “specific”. We changed “After subunit joining” to “Upon subunit joining”. We also included Pi release as part of the mechanism of conformational changes. The sentence now reads:

“Upon subunit joining IF2 hydrolyzes GTP and, concomitant with inorganic phosphate (Pi) release, changes conformation facilitating fMet-tRNA_i^{fMet} accommodation into the P site and transition of the 70S initiation complex (70S-IC) to an elongation-competent ribosome.”

- introduction: “large 50S subunit” should better say large subunit (50S)

Fixed.

- lines 83 & 128: “transitions” doesn’t tell what is happening, in fact it back rotates

Old line 83 changed to: “...and the ribosome takes the non-rotated conformation.”

Old line 128 changed to: “**Compact IF2-GDP is bound to the non-rotated 70S-IC**”

- line 90: “hands tRNA into P-site” is incorrect concept (see comments on abstract), better would be accommodates

Fixed.

Furthermore, to be consistent throughout the text, we changed the title to:

“Compact IF2 allows initiator tRNA accommodation into the P site and gates the ribosome to elongation”

- lines 108/109: those two states appear similar to those described in the Myasnikov et al. NSMB 2005 paper (suggested to be GTP and GDP states at the time), is that the case? Maybe worth discussing there

Thank you for this suggestion. We have added the following statement to the text:

“The two class averages appear to be reminiscent of the low-resolution cryo-EM reconstructions of the *Thermus thermophilus* ribosome bound to IF2, in which the 30S subunit is rotated in the 70S-IC bound to IF2-GDPCP and non-rotated upon binding IF2-GDP⁹.”

In response to reviewer 2 suggestion, we added to the discussion section a paragraph drawing a perspective regarding the extended IF2-GDP reported by Myasnikov et al. NSMB 2005 in the *T. thermophilus* 70S-IC and our compact IF2-GDP in the *P. aeruginosa* 70S-IC. It reads as follows:

“In the previous low-resolution cryo-EM structure of the *T. thermophilus* 70S-IC, IF2-GDP was observed in the extended conformation, which may seem at odd with the compact conformation of IF2-GDP reported here. A possible explanation could be attributed to the presence or absence of P_i in IF2-GDP. However, in our structures, it is clear that GDP is bound without P_i . Similarly, the previous *T. th* IF2-GDP-70S-IC was prepared using GDP⁹, and it is therefore unlikely that unreleased P_i is bound to IF2. Another possibility considers the ribosome as a Brownian nanomachine, exploiting thermal motions for many of the conformational changes during protein synthesis⁴⁰. It is conceivable that thermal motions of the ribosome could play an active role in promoting conformational changes of IF2. Both the *T. th* IF2-GDP-70S-IC and our *P. ae* 70S-IC were prepared by incubating the complexes at 37°C (see ref. ⁹ and Methods), which corresponds to the temperature at which *P. aeruginosa* ribosomes operate but is substantially lower than the optimum temperature for *T. thermophilus* (~75°C). Therefore, a reduced flexibility at low temperature of the thermophilic ribosomes could have prevented IF2 from assuming the compact configuration.”

- as the structure sorting and refinements heavily rely on focused classifications and refinements it may be worth mentioning some key references such as Curr Op Struct Biol 2017, eLife 2018, Biosci Rep 2018 etc.

Thank you for this suggestion. We added the following three references:

23. von Loeffelholz, O. et al. Focused classification and refinement in high-resolution cryo-EM structural analysis of ribosome complexes. *Curr. Opin. Struct. Biol.* **46**, 140-148 (2017).
24. Serna, M. Hands on Methods for High Resolution Cryo-Electron Microscopy Structures of Heterogeneous Macromolecular Complexes. *Front. Mol. Biosci.* **6**, 33 (2019).
25. Punjani, A. & Fleet, D.J. 3D variability analysis: Resolving continuous flexibility and discrete heterogeneity from single particle cryo-EM. *J. Struct. Biol.* **213**, 107702 (2021).

- line 129: if IF2 is resolved it is not clear why additional focused refinement was needed, maybe reformulate

To clarify this point, we modified the text to:

“The non-rotated ribosome particles were sorted on the basis of the presence of continuous EM density for compact IF2...”

- focused refinement with signal subtraction: did this include re-centring? that could improve image processing even further

Interesting and thank you for the suggestion. The local refinements presented in the manuscript did not include re-centering. We re-ran the local refinement jobs including this option. It did not, however, further improve the quality of the maps nor the resolution.

- 155: what is the issue with the poorly defined linker, does it affect the conformation or the conclusions?

We agree that the use of “poorly defined” to describe the density of the linker brings confusion. The aim was to let the reader know that the density of the C1-C2 linker is not as well-defined as for the remaining of IF2. Yet, as shown in Fig. 4c, the density is decent enough to be more affirmative here. Therefore, we changed to text to:

“The C2-domain docks against domain C1 (Figs. 1b and 4b; Supplementary Fig. 7), and Arg⁷⁴² in the C1-C2 linker region engages in electrostatic interactions with a negative patch lining one side of helix H12 in domain C1 (Fig. 4c).”

For consistency, we correspondingly modified the legend of Fig. 4c.

- line 165: sw2 alpha helix: this is rather buried next to the active site in the G domain, what does the rotation imply for neighbouring secondary structure elements and side-chains of IF2 (once it has left the SRL region)?

The suggestion made by Reviewer 1 allows to better illustrate this point. We now show Fig. 3 panels e and f in the same orientation. Furthermore, we made a new Supplementary Fig. 10 which shows how both switch regions 1 and 2 stabilize domain C1 in the extended IF2-GDPCP. In compact IF2-GDP, the sw2 alpha helix rotation by $\sim 65^\circ$ is accompanied by relocation of domain C1. Fig. 3e shows that in IF2-GDP, sw2 contacts helix H10 of domain C1. In extended IF2-GDPCP (Fig. 3f), sw2 helix contacts helix H9 located on the opposite side of domain C1. Detailed interactions with EM density for sw2 and H10 of domain C1 in IF2-GDP, and sw2 and H9 of domain C1 in IF2-GDPCP are shown in Fig. 4e and Supplementary Fig. 10c,e, respectively.

- line 168: “GTP hydrolysis must occur” and also Pi release. Mechanistically this is an important step

You are correct. We changed the text to:

“...indicating that GTP hydrolysis and P_i release must occur for IF2 to transition to the compact conformation on the ribosome.”

- buried surface: the change is about 10%, which not sound that large/significant. Maybe better to focus on the loss of specific interactions with the helix 6 region etc.

We agree that ~10% reduction in the total buried surface may seem insignificant. However, the reduction of contact area is the result of IF2 dissociation from the 30S subunit and the SRL, which in that regards, makes it significant.

We have edited the text to make clear that the lost contacts are from the 30S and the SRL. The text now reads:

“The movement of compact IF2 in structure I-B increases the gap between the 50S subunit and the G-domain by ~4Å around the SRL, and the displacement of domain II away from the 30S subunit combined with further back-rotation of the 30S subunit, from ~1.4° in structure I-A to ~0.7° in structure I-B, eliminate interactions with the small subunit (Fig. 6a, b). Correspondingly, the total buried surface area by compact IF2-GDP is reduced from ~2616 Å² in structure I-A to ~2409 Å² in structure I-B, consistent with weaker interactions between ribosome I-B and IF2-GDP primed to dissociate from the 70S-IC^{9,15}.”

- line 239 and following paragraph: the idea that the N-domain holds on the 30S and promotes subunit joining with the 50S was originally described in reference 18 (could be mentioned from the beginning). The fact that additional interactions provided by the N-domain anchor IF2 on the ribosome is consistent with observations made in ref. 18. Interesting that the interaction is released upon back-rotation of the 30S subunit

You are correct and thank you for noticing this. The text has been modified accordingly and this reference is now cited at the beginning of the section.

“The N-domain of IF2 is known to promote stable joining of the 30S and 50S subunits to form the 70S-IC²¹.”

- line 282: how does the crystal structure of IF2 (Eiler et al, at the time suggested to be in a different conformation than on the ribosome) now compare with the IF2 conformation seen in the compact state? Is shown in Suppl but not discussed

Thank you for the suggestion. We now include the following in the Discussion section:

“In the crystal structures of ribosome-free GDP-bound and apo IF2, domain C1 occupies different positions relative to domains I-II, and domain C2 is not visible likely due to intrinsic flexibility (Supplementary Fig. 13a)¹⁹. The sw2-helix in ribosome-free IF2-GDP is disordered probably because of the absence of interaction with domain C1. In the 70S-IC, the space restraints imposed by the uL11-stalk may facilitate the re-location of domain C1 in IF2-GDP such that it docks against the sw2-helix in the G-domain. The flexibility of helix H8 in IF2 allows it to bend and accommodate the new position of domain C1 under the uL11-stalk, a feature that was not observed in the ribosome-free crystal structures of IF2 (Supplementary Fig. 13b).”

- line 288: “we propose the following model for IF2-mediated translation initiation”: maybe add:

regarding the final steps of the process until IF2 is released (considering that IF2 and IF1/3 have a role in pre-initiation and early 30SIC formation etc.)

We have included this suggestion, thank you! Now the text reads:

“Based on our structures, we propose the following model for IF2-mediated translation initiation regarding the final steps of the process until IF2 is released.”

- 291/292: conformational changes of IF2 and subunit rotation appear to be induced by Pi release; the question is what starts the process: GTP hydrolysis is required but not sufficient, it needs Pi release (see published kinetics studies)

To address this point, we changed the text to: “Upon GTP hydrolysis and P_i release, the sw2-helix in the G-domain rotates and sw1 unfolds, releasing restraints on domain C1 and increasing its intrinsic flexibility.”

- angle estimation of 30S rotation: around 1° it may become difficult to have precise values. For the calculation, were the 50S subunit parts aligned to each other and then the angle measured on the 30S? (23 S rRNA is indicated, full 50S might be better, without the flexible parts). This gives more precise values and avoids getting averaged value effects with the 50S subunit not being well aligned

This you for this important suggestion. To measure the rotation angle of the 30S subunit in each complex presented, we now align the 50S subunit excluding flexible regions. In the Methods section, the text reads:

“The 30S subunit rotation was calculated in PyMOL by aligning the 50S subunit, excluding flexible regions (23S rRNA residues 2090-2174 of the uL1-stalk, residues 1045-1099 of the uL11-stalk, and ribosomal proteins uL10 and uL11), of the four 70S-IC structures (I-A, I-B, II-A, II-B) with that of the non-rotated *P. aeruginosa* 70S ribosome [PDB 6SPG⁴³] and the angle measured between phosphate atoms of C1478 located in helix h44 of 16S rRNA (considered the center of rotation) and U81 located in the foot region of the small subunit.”

- kink in the tip of helix 8: any particular amino acid sequence that would favour this (Gly etc.)?

There are no usual suspect residues such as Gly or Pro at the kink. We illustrate the residues involved at the kink in panel d of Fig. 4.

- “focused EM density” sounds imprecise, maybe better: cryo-EM density obtained after focused refinement (text and legends)

Fixed.

- “classes” should be better called class averages

Fixed.

- clashes in the atomic model should be checked for / refined (see validation report)

Fixed.

- cryo-EM maps, half maps, masks etc. and fully refined atomic models should be deposited in the EMDB and PDB, and representative data deposited onto the EMPIAR data base

We have deposited cryo-EM maps, half-maps, masks, and maps obtained after focused refinement in the EMDB. The EMDB accession numbers are:

EMD-26628 (70S, I-A), EMD-26629 (compact IF2-GDP from focused refinement, I-A), EMD-26630 (70S-IF2-GDP composite, I-A), EMD-26631 (70S, I-B), EMD-26632 (compact IF2-GDP from focused refinement, I-B), EMD-26633 (70S-IF2-GDP composite, I-B), EMD-26634 (70S-IF2-GDPCP, II-A), EMD-26635 (70S-IF2-GDPCP, II-B), and EMD-26553 (IF2 N2 sub-domain from focused refinement).

All models are deposited in the PDB with accession numbers:

7UNQ (compact IF2-GDP, I-A), 7UNR (70S-IF2-GDP composite, I-A), 7UNT (compact IF2-GDP, I-B), 7UNU (70S-IF2-GDP composite, I-B), 7UNV (70S-IF2-GDPCP, II-A), 7UNW (70S-IF2-GDPCP, II-B), and 7UIU (IF2 N2 sub-domain).

The raw data (unaligned cryo-EM multi-frame micrographs) has been deposited in the EMPIAR database under accession number EMPIAR-11012.

All accession numbers have been incorporated into the manuscript.

Thank you for the opportunity to review this work. Bruno Klaholz

REVIEWERS' COMMENTS

Reviewer #1 (Remarks to the Author):

The authors have fully addressed my points. There is only one minor issue left in lines 318-320: The description of the native initiation pathway should be slightly rephrased, terms like “incubated with” or “added to” seem inappropriate here. An alternative phrasing might be, for instance, “..., in which the 30S subunit first forms a complex with initiation factors IF1, IF2, IF3 and fMet-tRNA^{fMet}, followed by joining of the 50S subunit and GTP hydrolysis by IF2.”
Niels Fischer

Reviewer #2 (Remarks to the Author):

Overall, the authors have positively responded to my comments and have amended their text accordingly. However, prior to publication I have only one request. In lines 76-77 the authors state that domains C1 and C2 remain flexible. In my opinion, the article of Wienk, H., Tishchenko, E., Belardinelli, R. et al. (2012) Structural Dynamics of Bacterial Translation Initiation Factor IF2 J. Biol. Chem. 287, 10922–10932 should be cited at this point. In fact, the presence of a very long, rigid alpha helix connecting domains C1 and C2 in bacterial IF2, by analogy with the archaeal aIF5b, became common place and several functional models based on this mantra were proposed. However, the results of Wienk et al. did not support the levering model but demonstrated for the first time that, although partly helical, the connector of C1 and C2 is not a continuous, rigid alpha-helix and NMR relaxation analysis demonstrated that the two domains are not rigidly linked but can rotate and tumble independently from one another.

Reviewer #3 (Remarks to the Author):

All suggestions were taken into account.

Final suggestion: “sequence homology” should say sequence similarity.

REVIEWERS' COMMENTS (Final)

Reviewer #1 (Remarks to the Author):

The authors have fully addressed my points. There is only one minor issue left in lines 318-320: The description of the native initiation pathway should be slightly rephrased, terms like “incubated with” or “added to” seem inappropriate here. An alternative phrasing might be, for instance, “..., in which the 30S subunit first forms a complex with initiation factors IF1, IF2, IF3 and fMet-tRNA^{fMet}, followed by joining of the 50S subunit and GTP hydrolysis by IF2.” Niels Fischer

Thank you Dr. Fischer for this suggestion. The text now reads:

“Our preparation of the 70S-IC complex for cryo-EM structure determination deviates from the native initiation pathway, in which the 30S subunit first forms a complex with mRNA, initiation factors IF1, IF2, IF3 and fMet-tRNA^{fMet}, followed by joining of the 50S subunit and GTP hydrolysis by IF2.”

Reviewer #2 (Remarks to the Author):

Overall, the authors have positively responded to my comments and have amended their text accordingly. However, prior to publication I have only one request. In lines 76-77 the authors state that domains C1 and C2 remain flexible. In my opinion, the article of Wienk, H., Tishchenko, E., Belardinelli, R. et al. (2012) Structural Dynamics of Bacterial Translation Initiation Factor IF2 J. Biol. Chem. 287, 10922–10932 should be cited at this point. In fact, the presence of a very long, rigid alpha helix connecting domains C1 and C2 in bacterial IF2, by analogy with the archaeal aIF5b, became common place and several functional models based on this mantra were proposed. However, the results of Wienk et al. did not support the levering model but demonstrated for the first time that, although partly helical, the connector of C1 and C2 is not a continuous, rigid alpha-helix and NMR relaxation analysis demonstrated that the two domains are not rigidly linked but can rotate and tumble independently from one another.

We are thankful to this reviewer for catching this omission. We have added the suggested reference to the manuscript (reference 20).

Reviewer #3 (Remarks to the Author):

All suggestions were taken into account.

Final suggestion: “sequence homology” should say sequence similarity.

Thank you Dr. Klaholz, we completely agree. All instances of “sequence homology” have been changed to “sequence similarity”.